# Surface-associated antigen induces permeabilization of primary mouse B-cells and lysosome exocytosis facilitating antigen uptake and presentation to T-cells

Fernando Y Maeda[1†], Jurriaan JH van Haaren[1†], David B Langley[2], Daniel Christ[3], Norma W Andrews[1]*, Wenxia Song[1]*

[1]Department of Cell Biology and Molecular Genetics, University of Maryland, College Park, United States; [2]Immunology Division, Garvan Institute of Medical Research, Darlinghurst, Australia; [3]Immunology, Garvan Institute of Medical Research, Darlinghurst/Sydney, Australia

*For correspondence: andrewsn@umd.edu (NWA); wenxsong@umd.edu (WS)

[†]These authors contributed equally to this work

Competing interest: The authors declare that no competing interests exist.

**Abstract** B-cell receptor (BCR)-mediated antigen internalization and presentation are essential for humoral memory immune responses. Antigen encountered by B-cells is often tightly associated with the surface of pathogens and/or antigen-presenting cells. Internalization of such antigens requires myosin-mediated traction forces and extracellular release of lysosomal enzymes, but the mechanism triggering lysosomal exocytosis is unknown. Here, we show that BCR-mediated recognition of antigen tethered to beads, to planar lipid-bilayers or expressed on cell surfaces causes localized plasma membrane (PM) permeabilization, a process that requires BCR signaling and non-muscle myosin II activity. B-cell permeabilization triggers PM repair responses involving lysosomal exocytosis, and B-cells permeabilized by surface-associated antigen internalize more antigen than cells that remain intact. Higher affinity antigens cause more B-cell permeabilization and lysosomal exocytosis and are more efficiently presented to T-cells. Thus, PM permeabilization by surface-associated antigen triggers a lysosome-mediated B-cell resealing response, providing the extracellular hydrolases that facilitate antigen internalization and presentation.

## Editor's evaluation

The revisions have addressed all reviewer concerns including recovery of B cells that had undergone significant morphological change consistent with extensive plasma membrane permeabilization/lysis. Congratulations on the exciting study revealing an important role of plasma membrane permeabilization in antigen capture by B cells.

## Introduction

B-cells are responsible for generating antibody responses that neutralize pathogens and attract other immune cells. B-cell activation is initiated by the B-cell receptor (BCR), which surveys antigen through its membrane-anchored immunoglobulin (*Reth, 1994*). Antigen-BCR interaction induces signaling cascades and antigen internalization, followed by intracellular processing and surface presentation to T-cells. Antigen presentation is essential for the activation of B-cells and their differentiation into high-affinity memory or antibody-secreting cells (*Shlomchik and Weisel, 2012*). A property that is critical for maximizing humoral protection is the ability of clonal-specific BCRs to recognize antigens in their different physical, chemical, and biological forms.

 

Antigen encountered by B-cells in vivo is often tightly associated with the surface of pathogens, such as parasites, bacteria, and viruses, and/or antigen-presenting cells, such as follicular dendritic cells (*Gonzalez et al., 2011*). Internalization, processing, and presentation of such surface-bound antigens are essential for specific B-cells to obtain T-cell help, which is critical for B-cell activation and differentiation. Follicular dendritic cells, which are uniquely present in germinal centers of secondary lymphoid organs, internalize antigens that drain into these organs and present them to B-cells (*Suzuki et al., 2009*; *Cyster, 2010*). Competition between high and low-affinity B-cells to acquire antigen from follicular dendritic cells is a critical step in the selection of high-affinity cells that differentiate into memory B-cells and long-lived plasma cells.

B-cells, follicular B-cells in particular, are thought to have a limited ability to phagocytose large insoluble antigen particles (*Vidard et al., 1996*). However, B-cells are able to extract and endocytose antigen that is tightly associated with non-internalizable surfaces (*Batista and Neuberger, 2000*). Importantly, the efficiency of antigen presentation by B-cells appears to depend more strongly on the BCR-antigen binding affinity when the antigen is associated with non-internalizable surfaces, compared to antigen bound to internalizable particles (*Batista and Neuberger, 2000*). Recent studies using antigen-coated beads, planar lipid bilayers, or plasma membrane (PM) sheets revealed two major mechanisms by which B-cells extract antigen from non-internalizable surfaces for endocytosis. Mechanical forces, generated by non-muscle myosin II (NMII) activation at sites of antigen-BCR inter-action, can directly pull antigen from presenting surfaces for endocytosis. When mechanical forces alone are not sufficient, hydrolases released from lysosomes cleave surface-associated antigen to facilitate internalization (*Yuseff et al., 2011*; *Natkanski et al., 2013*; *Spillane and Tolar, 2017*; *Wang et al., 2018b*). Surface-associated antigen was previously shown to induce polarization of B-cell lyso-somes towards antigen-binding sites (*Yuseff et al., 2011*), but the mechanism responsible for trig-gering lysosome exocytosis and release of hydrolytic enzymes was unknown.

When cells are permeabilized by physical tearing or pore-forming proteins, $Ca^{2+}$ influx triggers rapid exocytosis of lysosomes as part of the process that repairs the PM and prevents cell death (*Reddy et al., 2001*; *Andrews et al., 2014*). Since its discovery several decades ago (*Rodríguez et al., 1997*), $Ca^{2+}$-dependent exocytosis of lysosomes has been observed in many cell types (*Zhang et al., 2007*; *Naegeli et al., 2017*; *Villeneuve et al., 2018*; *Ibata et al., 2019*). We previously reported that permeabilization of the PM of mouse splenic B-cells with the pore-forming toxin streptolysin O (SLO) triggers lysosomal exocytosis, releasing hydrolases extracellularly and exposing the luminal epitope of the lysosome-associated protein LIMP-2 on the cell surface. B-cells rapidly reseal these PM lesions in a process that requires lysosomal exocytosis (*Miller et al., 2015*). Surprisingly, in this study, we found that interaction of the BCR with surface-associated antigen can permeabilize the mouse primary B-cell PM, triggering a resealing mechanism that involves exocytosis of lysosomes. We investigated this process by determining if antigen-induced PM permeabilization depends on the BCR-antigen-binding affinity, BCR signaling and NMII motor activity, and if it influences the ability of B-cells to internalize and present surface-associated antigens to T-cells.

## Results

### BCR interaction with surface-associated antigen induces B-cell PM permeabilization at antigen-binding sites

We initially utilized two experimental models previously used to study BCR-mediated internal-ization of surface-associated antigen: F(ab')₂-anti-mouse IgM+ G (αM, which binds and activates mouse BCRs) immobilized on beads or tethered to planar lipid bilayers (PLB) by biotin-streptavidin interaction. Beads or PLB coated with transferrin (Tf) at similar surface density as αM were used as controls, as Tf does not activate the BCR and interacts with the Tf receptor with similar affinity as the *bona fide* antigen hen egg lysozyme (HEL) binds to the BCR of transgenic MD4 mouse B-cells (*Batista and Neuberger, 1998*; *Fuchs and Gessner, 2002*). Strikingly, live imaging revealed influx of the membrane-impermeable dye propidium iodide (PI) at sites of mouse splenic B-cell contact with αM-beads, indicating that PM permeabilization occurred at bead-binding locations (*Figure 1A*, *Figure 1—figure supplement 1* and *Videos 1–3*). While similar percentages of B-cells bound αM- or Tf-beads (*Figure 1B*), a significantly higher fraction of B-cells binding αM-beads became PI-positive (*Figure 1C*). Flow cytometry analysis confirmed the increased PI entry in B-cells binding αM-beads

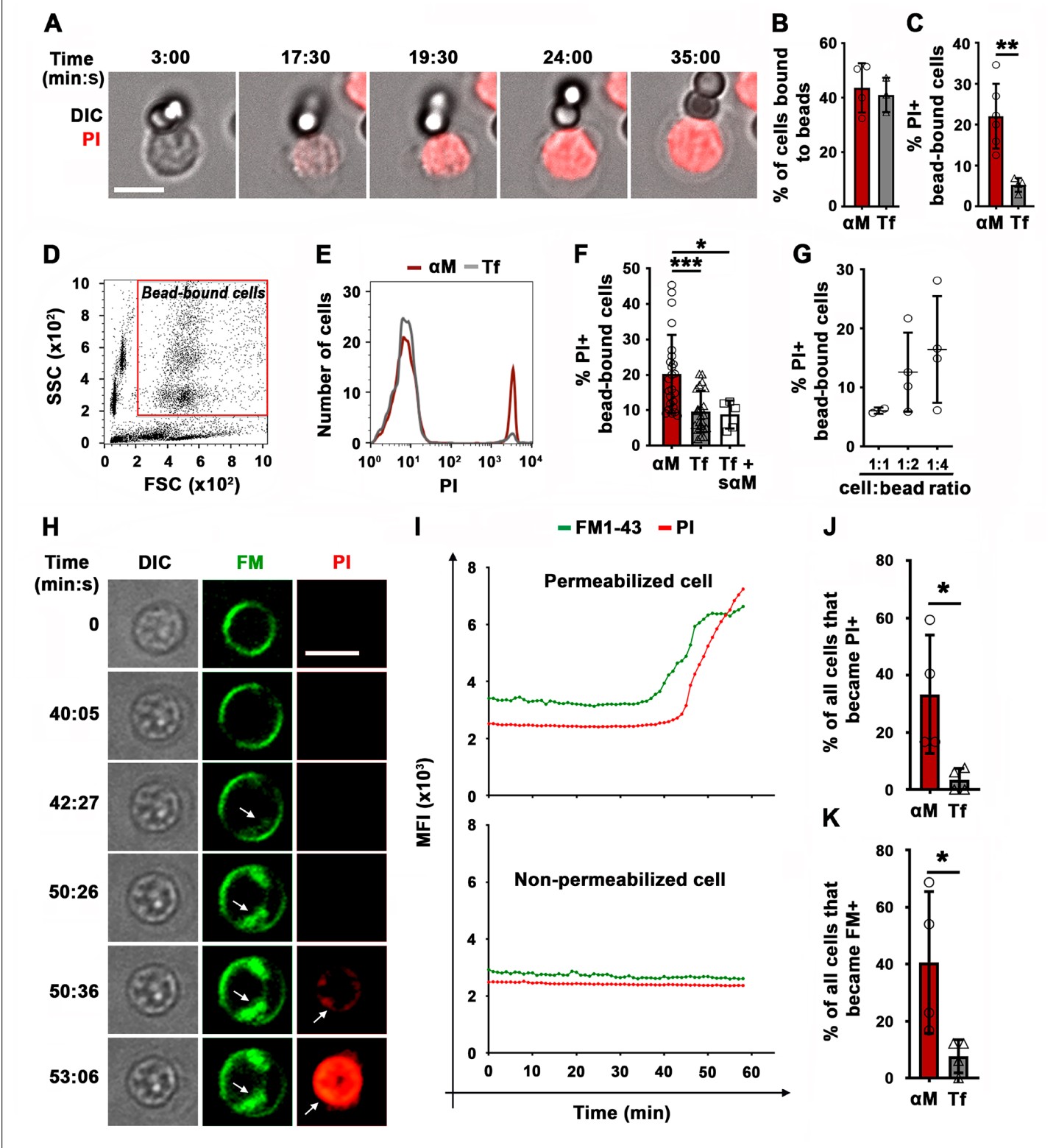

**Figure 1.** BCR binding to surface-associated ligands causes B-cell PM permeabilization. (**A**) Time-lapse images of a splenic B-cell incubated with αM-beads (1:2 cell:bead ratio) in the presence of PI (***Video 1***). (**B**) Percentages of B-cells bound to beads. (**C**) Percentages of PI-positive (PI+) cells in bead-bound B-cells at 30 min. (**D**) Gate for bead-bound B-cells in forward and side scatter flow cytometry dot plot. (**E**) Histograms of PI fluorescence intensity (FI) of αM- and Tf-bead-bound B-cells after 30 min incubation, showing 1000 cells per condition. (**F**) Percentages of PI+ bead-bound B-cells after 30 min incubation with αM- or Tf-beads with or without soluble αM (sαM). (**G**) Percentages of PI+ bead-bound B-cells after 30 min at indicated

*Figure 1 continued on next page*

*Figure 1 continued*

cell:αM bead ratios. (**H**) Time-lapse images of a B-cell interacting with αM-PLB in the presence of FM1-43 and PI (arrows, FM1-43 or PI entry, *Video 4*). (**I**) Mean fluorescence intensity of FM1-43 (green lines) and PI (red lines) in a defined intracellular region of a permeabilized (top) and non-permeabilized (bottom) cell over time. (**J**) Percentages of PI+ B cells interacting with αM- or Tf-PLB for 60 min. (**K**) Percentages of B-cells interacting with αM- or Tf-PLB for 30 min showing intracellular FM staining (FM+). Data points represent independent experiments (mean ± SD) (**B, C, F, G, J, K**). Bars, 5 µm. *p ≤ 0.05, **p ≤ 0.01, ***p ≤ 0.005, unpaired Student's *t*-test (**B, C, J, K**) or one-way ANOVA (**F**).

The online version of this article includes the following figure supplement(s) for figure 1:

**Figure supplement 1.** BCR binding to αM-beads causes localized PM permeabilization in B-cells.

**Figure supplement 2.** Identification of bead-bound B-cells by flow cytometry.

**Figure supplement 3.** BCR binding to αM-beads does not increase apoptosis in B-cells.

**Figure supplement 4.** Sudden increases in intracellular staining with the lipophilic FM dye in B-cells permeabilized by interaction with αM-PLB.

**Figure supplement 5.** The lipophilic FM dye enters B-cells permeabilized by αM-PLB and stains the nuclear envelope.

**Figure supplement 6.** BCR cross-linking with soluble ligands does not permeabilize B-cells but induces a punctate form of FM uptake at the cell periphery that is distinct from the massive FM influx induced by surface-associated ligands.

when compared to Tf-beads (*Figure 1D–G* and *Figure 1—figure supplement 2*). Addition of soluble F(ab')$_2$-anti-mouse IgM+ G (sαM, also capable of binding and activating the BCR) did not increase the frequency of PI entry in B-cells binding to Tf-beads (*Figure 1F*). The percentage of cells positive for cleaved caspase-3, an early apoptotic marker, was similar in B-cells interacting or not with αM- or Tf-beads and only increased significantly after treatment with staurosporine (*Figure 1—figure supplement 3*), suggesting that PM permeabilization is not associated with apoptosis. Similar observations were made using the PLB system that allows lateral movement of the tethered antigen (*Dustin et al., 2007*). Significantly more B-cells became PI-positive when contacting αM-PLB when compared to Tf-PLB (*Figure 1H–J*).

PM permeabilization in B-cells binding to αM-PLB was also observed using membrane-impermeable lipophilic FM probes. These fluorescent dyes have been used extensively to assess PM integrity, because they only label the outer PM leaflet of intact cells but rapidly stain intracellular membranes when entering the cytosol (*Bansal et al., 2003*; *McNeil et al., 2003*; *Demonbreun et al., 2019*). After >30 min of interaction with αM-PLB, we observed sudden, massive increases in FM1-43 staining of intracellular membranes, including the nuclear envelope (*Figure 1H and I*, *Figure 1—figure supplements 4 and 5* and *Video 4*). Consistent with the PI entry results (*Figure 1J*), significantly more B-cells showed a sudden increase in intracellular FM staining when contacting αM-PLB compared to Tf-PLB (*Figure 1K*). This characteristic pattern of sudden FM influx with staining of the nuclear envelope was only observed in B-cells that eventually became PI-positive, not in cells that remained PI-negative

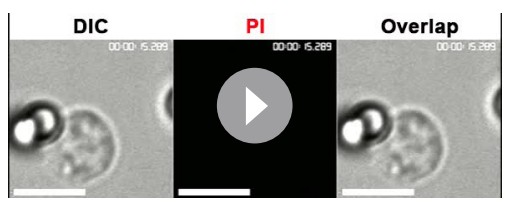

**Video 1.** BCR binding to αM-beads permeabilizes the PM of splenic B-cells. Splenic B-cells were incubated with αM-beads at 4 °C and warmed to 37 °C in a live imaging chamber with 5 % CO$_2$ in DMEM-BSA. Time-lapse images were acquired for 60 min at one frame/15 s in the presence of PI (red) using a spinning disk fluorescence microscope (UltraVIEW VoX, PerkinElmer with a 63 × 1.4 N.A. oil objective). The arrow indicates the moment of PI entry. Time is displayed as hour: minutes: seconds. The video is displayed at 20 frames/s. Bar, 5 µm.

https://elifesciences.org/articles/66984/figures#video1

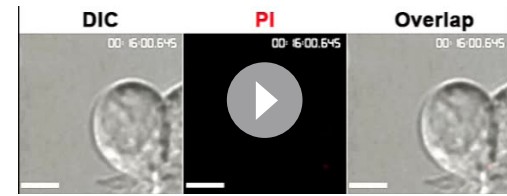

**Video 2.** BCR binding to αM-beads causes localized PM permeabilization in A20 B-cells (cell line). A20 B-cells were incubated with αM-beads in a live imaging chamber at 37 °C with 5 % CO$_2$ in DMEM/BSA. Time-lapse images were acquired for 65 min at one frame/20 s in the presence of PI using a spinning disk fluorescence microscope (UltraVIEW VoX, PerkinElmer with a 63 × 1.4 N.A. oil objective). The arrow points to the beads and the arrowhead points to the site of entry and subsequent flow of PI into the cell. Beads appear red as a result of autofluorescence. Time is displayed as hour: minutes: seconds. The video is displayed at 20 frames/s. Bar, 5 µm.

https://elifesciences.org/articles/66984/figures#video2

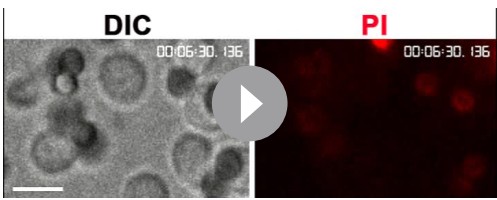

**Video 3.** Bead exchange between B-cells causes PM permeabilization. Splenic B-cells were incubated with αM-beads in a live imaging chamber at 37 °C with 5 % $CO_2$ in DMEM-BSA. Images were acquired for 60 min at one frame/30 s in the presence of PI using a spinning disk fluorescence microscope (UltraVIEW VoX, PerkinElmer with a 63 × 1.4 N.A. oil objective). The arrow points to the bead that was exchanged between cells (#1 and #2) and caused permeabilization of cell #2. Beads appear red as a result of autofluorescence. Time is displayed as hour: minutes: seconds. The video is displayed at 10 frames/s. Bar, 5 µm.
https://elifesciences.org/articles/66984/figures#video3

As an independent method to demonstrate antigen-induced permeabilization of B-cells, we took advantage of the ability of membrane-impermeable Ponceau 4 R to quench cytosolic fluorophores upon entering cells (*Tay et al., 2019*). Instead of monitoring nuclear or intracellular membrane staining by membrane-impermeable fluorescent dyes, we determined the percentage of B-cells

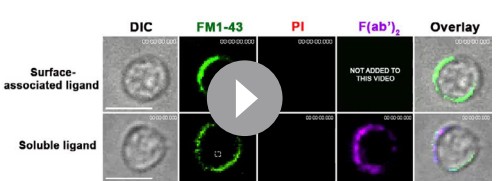

**Video 4.** Surface-associated ligand induces B-cell permeabilization and massive FM influx, while soluble ligand does not cause permeabilization but induces endocytosis, detected as puncta at the cell periphery. Top: B-cells pre-labeled with FM1-43 (green) were added to αM-PLB (surface-associated ligand). Bottom: B-cells pre-labeled with FM1-43 (green) and anti-BCR antibodies followed by secondary fluorochrome-labeled crosslinking antibodies (magenta) (soluble ligand). Under both conditions, cells were imaged at 37 °C in the presence of FM1-43 (green), and PI (red) was added to detect PM permeabilization. Images were acquired for 60 min at one frame/30 s or 15 s using a spinning disk fluorescence microscope (UltraVIEW VoX, PerkinElmer with a 60 × 1.4 N.A. oil objective). Time is displayed as minutes: seconds after cells contacted αM-PLB. The white box indicates the intracellular area used to measure FI levels of intracellular FM1-43 (see Figure 1I and Figure 1—figure supplement 4). The arrow indicates the massive influx of FM1-43 in cells permeabilized during contact with αM-PLB. The arrowheads indicate areas where peripheral FM1-43 puncta (likely endosomes) were observed next to clusters of crosslinked BCR (magenta). The video is displayed at 20 frames/s. Bar, 5 µm.
https://elifesciences.org/articles/66984/figures#video4

during interaction with αM-PLB (*Figure 1—figure supplement 4*). Since FM lipophilic dyes can also be internalized through surface receptor endocytosis, we activated BCR endocytosis by cross-linking surface BCRs using soluble F(ab')₂ goat-anti-mouse IgM+ G antibodies followed by fluorescent F(ab')₂ anti-goat-IgG (*Song et al., 1995*; *Hoogeboom and Tolar, 2016*). Under these conditions, which did not cause PM permeabilization, we observed FM1-43 uptake appearing as small peripheral puncta that colocalized with BCR cross-linking antibodies. Such endosome-associated FM1-43 staining pattern was markedly different from the sudden, massive FM influx observed shortly before PI entry in permeabilized cells (*Figure 1—figure supplement 6* and *Video 4*). Collectively, these data show that the sudden, massive influx of FM dyes during αM-PLB binding is caused by B-cell permeabilization, and not by a gradual endocytosis of the PM-associated tracer triggered by BCR engagement.

pre-loaded with carboxyfluorescein succinimidyl ester (CFSE) that lost their cytosolic fluorescence as a consequence of Ponceau 4 R entry during PM permeabilization. To validate this method, we first permeabilized B-cells with the pore-forming toxin streptolysin O (SLO). In the presence of Ponceau 4 R, the percentage of B-cells with reduced CFSE fluorescence increased significantly after exposure to SLO (*Figure 2A and B*), mimicking what we previously observed for PI entry in SLO-treated B-cells (*Miller et al., 2015*). Thus, quenching of cytoplasmic CFSE by the membrane-impermeable Ponceau 4 R is a potent indicator of PM permeabilization. Using this method, we compared B-cells incubated with αM- or Tf-PLB by live imaging. A significantly higher fraction of CFSE-labeled B-cells showed fluorescence quenching when interacting with αM-PLB, quantified as the percentage of cells that lost >70% of their initial CFSE fluorescence (*Figure 2C and D* and *Video 5*). The average time for detection of αM-PLB-induced B-cell permeabilization measured by this quenching method was similar to what was observed for FM entry, while the average time for intracellular detection of PI showed a ~ 8 min delay (*Figure 2E*, *Figure 1—figure supplement 4* and *Video 4*). An analysis of the cumulative rate of influx of the three distinct tracers confirmed the small delay in PI detection (*Figure 2F*). Thus, FM influx and

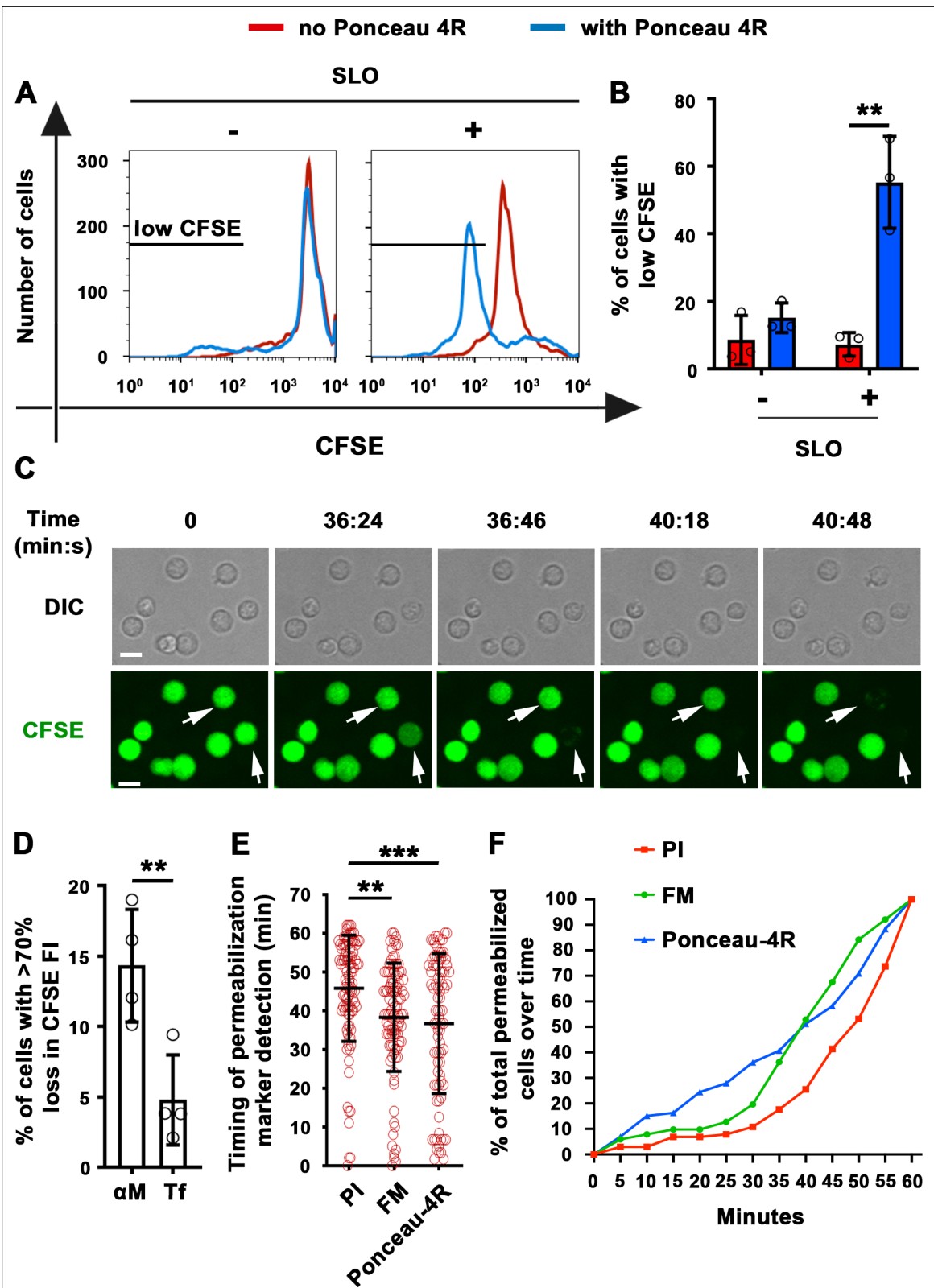

**Figure 2.** Extracellular Ponceau 4 R quenches cytoplasmic CFSE in αM-PLB-permeabilized B-cells. (**A**) Flow cytometry histograms of CFSE FI in B-cells incubated with or without SLO for 10 min in the presence or absence of Ponceau 4 R, showing 8500 cells per condition. (**B**) Percentages of cells with reduced CFSE in the presence or absence of Ponceau 4 R after treatment with or without SLO. Data points represent independent experiments (mean ± SD). (**C**) Time-lapse images of B-cells pre-stained with CFSE interacting with αM-PLB in the presence of Ponceau 4 R (arrows, cells with Ponceau

*Figure 2 continued on next page*

*Figure 2 continued*

4 R quenching of cytoplasmic CFSE) (***Video 5***). (**D**) Percentages of B-cells with more than 70 % loss of CFSE FI after 60 min interaction with αM- or Tf-PLB. Data points represent independent experiments (mean ± SD). (**E**) Timing of PI, FM1-43 entry or Ponceau 4R-mediated CFSE quenching in B-cells interacting with αM-PLB. Data points represent individual cells in at least four independent experiments (mean ± SD). (**F**) Cumulative percentages of total permeabilized B-cells detected over time in four independent experiments. Bars, 5 μm. **p ≤ 0.01, ***p ≤ 0.005, unpaired Student's *t*-test (**B, D**) or one-way ANOVA (**E**).

Ponceau 4R-mediated quenching are more sensitive methods for detecting the onset of B-cell PM permeabilization when compared to PI influx, which is only clearly visualized after intercalation into double-stranded DNA inside the nucleus. Based on consistent results obtained with three different methods, we conclude that BCR binding to αM-coated surfaces (but not to soluble αM) causes localized permeabilization of the B-cell PM.

We next determined whether HEL, a bona fide antigen recognized by the BCR from MD4 mice, also caused B-cell permeabilization when tethered to artificial surfaces or presented as an integral membrane protein (mHEL) on the surface of live cells (*Batista et al., 2001*). Flow cytometry analysis revealed that similar fractions of MD4 B-cells become PI-positive after binding beads coupled to αM or to HEL (*Figure 3A–C*). In contrast, WT B-cells binding to HEL- beads showed a low percentage of PI-positive cells, similar to what is observed with Tf-beads (*Figures 1F and 3A–C*). Importantly, trans-membrane mHEL-GFP expressed on the surface of live COS-7 cells co-clustered with the BCR at sites of interaction with MD4 B-cells, followed by PI influx. This dramatic clustering pattern followed by permeabilization was not observed in WT B-cells, whose BCR is incapable of specifically recognizing HEL (*Figure 3D* and *Videos 6 and 7*). A significantly higher percentage of MD4 B-cells showed PI influx after interaction with COS-7 cells expressing mHEL-GFP, when compared to WT B-cells (*Figure 3E*). The percentage of PI-positive MD4 B-cells was also significantly higher after incubation with mHEL-expressing COS-7 cells than with mock-transfected cells (*Figure 3F*). Collectively, these results show that BCR binding to surface-associated antigen can cause permeabilization of the B-cell PM.

## Antigen-induced B-cell permeabilization requires high-affinity BCR-antigen binding, BCR signaling, and NMII motor activity

High-affinity binding of the BCR to antigen associated with non-internalizable surfaces induces high levels of BCR signaling, cytoskeleton reorganization, and antigen endocytosis (*Batista and Neuberger, 1998*; *Batista and Neuberger, 2000*; *Fleire et al., 2006*). To examine the impact of the BCR-binding affinity on antigen-induced PM permeabilization, we incubated MD4 B-cells with beads coated with equal densities of HEL or the duck egg lysozyme isoform DEL-I. The MD4 BCR binds DEL-I with >100 fold lower affinity than it binds HEL (*Langley et al., 2017*). As expected, the percentage of B-cells binding multiple beads was reduced when the BCR-antigen affinity decreased (*Figure 4—figure supplement 1A*), but B-cells binding one single bead were detected for both HEL and DEL-I and also Tf (*Figure 4—figure supplement 1*). In these single bead-bound populations, DEL-I-beads caused significantly less PI entry than HEL-beads (*Figure 4A*). Inhibition of signaling with the Src kinase inhibitor PP2 (*Cheng et al., 2001*) (iSrc) or the Bruton's Tyrosine Kinase inhibitor AVL-292 (*Aalipour and Advani, 2013*) (iBTK) (*Figure 4B and C*) also reduced PI entry in cells binding HEL-beads (*Figure 4D*). After contact with αM-PLB or αM-beads but not Tf-PLB or Tf-beads, surface BCRs became polarized toward PLB- or bead-binding sites within ~10 min, a period markedly shorter than what is required for detection of PM permeabilization through FM influx (*Figure 4E–H*, *Figure 4—figure supplement 2*, and *Video 8*). Importantly, the activated form of the actin motor

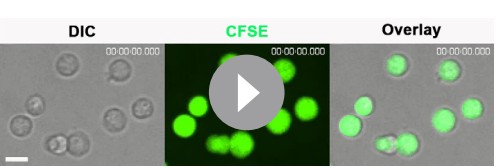

**Video 5.** B-cell PM permeabilization during binding to αM-PLB enables membrane-impermeable Ponceau 4 R to quench cytoplasmic CSFE fluorescence. Splenic B cells pre-labeled with CFSE in the cytosol were added to αM-PLB in a live imaging chamber at 37 °C with 5 % CO₂ in DMEM/BSA. Images were acquired for 60 min at one frame/10 s in the presence of Ponceau 4 R using a spinning disk fluorescence microscope (UltraVIEW VoX, PerkinElmer with a 40 × 1.4 N.A. oil objective). The arrow indicates CFSE-labeled B-cells that lost their cytosolic fluorescence as a result of PM permeabilization and Ponceau 4 R influx. Time is displayed as hour: minutes: seconds. The video is displayed at 30 frames/s. Bar, 5 μm.
https://elifesciences.org/articles/66984/figures#video5

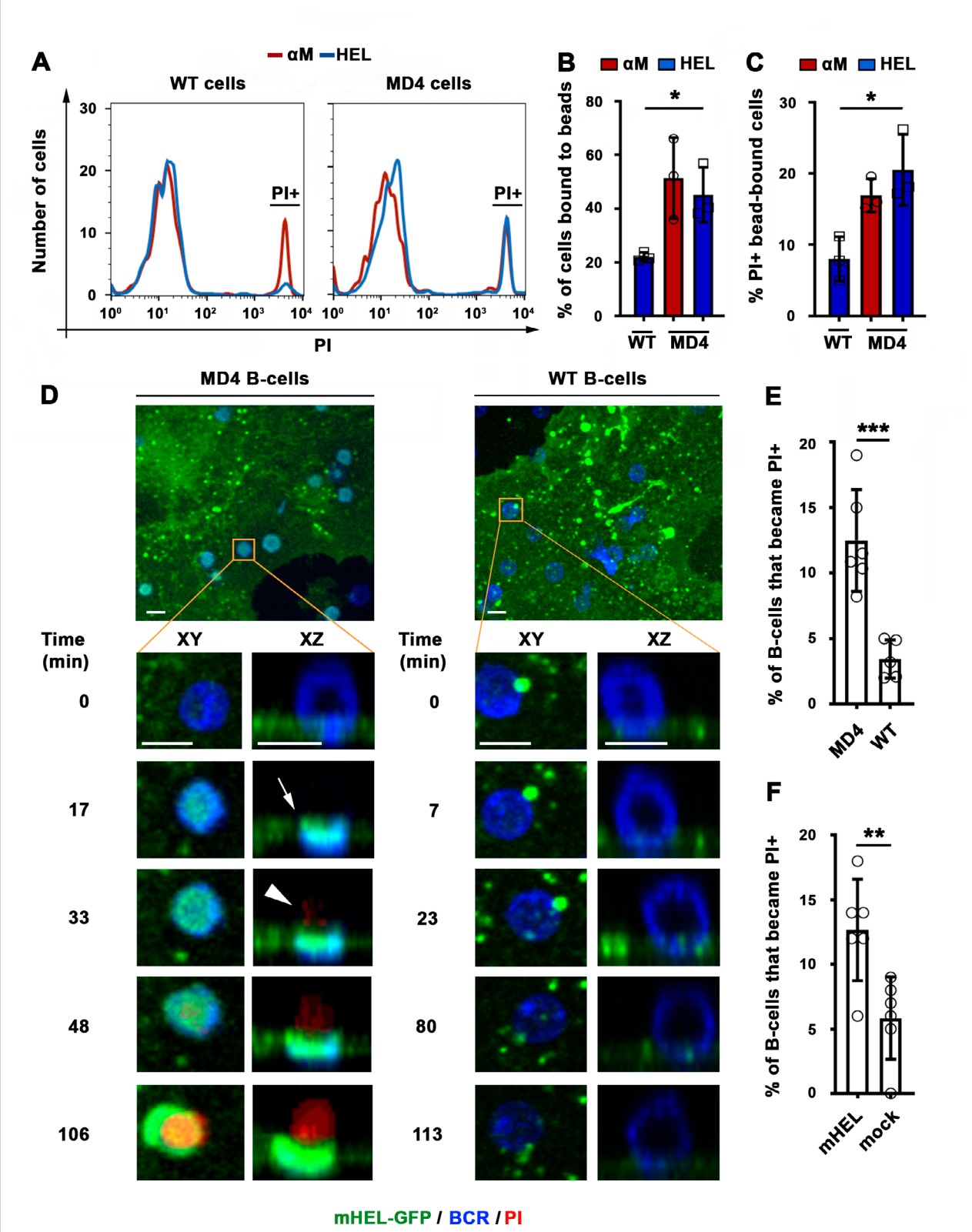

**Figure 3.** BCR-mediated binding of HEL coupled to beads or expressed as a transmembrane protein on COS-7 cells causes B-cell PM permeabilization. (**A**) Flow cytometry histograms of PI FI in WT or MD4 B-cells incubated with αM- or HEL-beads for 30 min by flow cytometry, showing 1000 cells per condition. (**B**) Percentages of WT and MD4 B-cells binding αM- or HEL-beads. Data points represent independent experiments (mean ± SD). (**C**) Percentages of PI+ bead-bound WT or MD4 B-cells after 30 min incubation. Data points represent independent experiments (mean ± SD). (**D**) Spinning

*Figure 3 continued on next page*

*Figure 3 continued*

disk time-lapse images of a MD4 B-cell (left panels) and a WT B-cell (right panels) interacting with a mHEL-GFP-expressing COS-7 cell in the presence of PI (*Videos 6 and 7*). Arrows, clustering of mHEL-GFP during B-cell binding; arrowheads, PI entry in the B-cell. (**E**) Percentages of PI+ MD4 and WT B-cells interacting with COS-7 cells transfected with mHEL-GFP. (**F**) Percentages of PI+ MD4 B-cells interacting with COS-7 cells transfected with mHEL-GFP or mock-transfected. Data points (**E and F**) represent individual videos from three to four independent experiments (mean ± SD). Bars, 5 µm *p ≤ 0.05, **p ≤ 0.01, ***p ≤ 0.005, unpaired Student's *t*-test (**E, F**) or one-way ANOVA (**B, C**).

protein NMII, detected through its phosphorylated light chain (pMLC), accumulated along with the BCR at αM-bead-binding sites (*Figure 4G*, *Figure 4—figure supplement 3* and *Video 9*). The fluorescence intensity ratios (FIR) of surface BCRs and pMLC were significantly higher in B-cells binding αM-beads than in cells binding Tf-beads (*Figure 4H and I*). Notably, inhibition of NMII motor activity with blebbistatin (Bleb) markedly reduced the number of B-cells that became PI-positive during interaction with αM-beads, without affecting the cells' ability to bind the beads (*Figure 4J and K*). Live imaging detected PI entry following a 'tug-of-war' between two B-cells simultaneously engaging an αM-bead (*Video 3* and *Figure 1—figure supplement 1C*), further supporting a role for NMII-mediated traction forces in antigen-induced PM permeabilization. Thus, our results indicate that PM permeabilization caused by surface-associated antigen requires strong BCR-antigen interaction and the subsequent activation of signaling and NMII motor activity.

## Antigen-induced B-cell permeabilization triggers lysosomal exocytosis as a PM repair response

Permeabilization with the pore-forming toxin SLO triggers exocytosis of lysosomes in mouse primary B-cells (*Miller et al., 2015*), a response to Ca²⁺ influx that is observed in several cell types and is

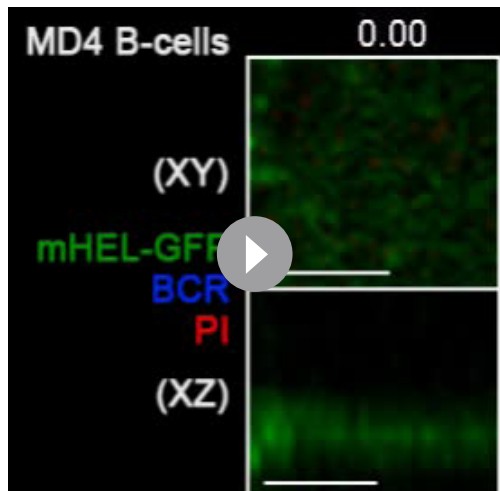

**Video 6.** Binding of MD4 B-cells to COS-7 cells expressing surface mHEL-GFP induces antigen clustering and PM permeabilization at interaction sites. MD4 splenic B-cells were incubated with mHEL-GFP-expressing COS-7 cells cultured on fibronectin-coated coverslips at 37 °C with 5 % CO₂ in DMEM/BSA. Images were acquired for 120 min at one frame/20 s in the presence of PI using a spinning disk fluorescence microscope (UltraVIEW VoX, PerkinElmer with a 40 × 1.3 N.A. oil objective). Shown are representative videos of XY (top) and XZ (bottom) views showing clustering of mHEL-GFP (arrows) and the intracellular influx of PI (arrowheads) at cell interacting sites. Time is displayed as minutes: seconds after the cell contacted the mHEL-GFP expressing COS cell. The video is displayed at 15 frames/s. Bar, 5 µm.

https://elifesciences.org/articles/66984/figures#video6

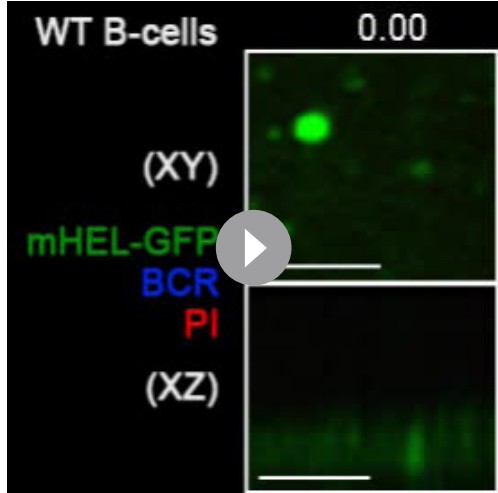

**Video 7.** Binding of WT B-cells to COS-7 cells expressing surface mHEL-GFP does not induce antigen clustering and PM permeabilization at interaction sites. WT splenic B-cells were incubated with mHEL-GFP-expressing COS-7 cells cultured on fibronectin-coated coverslips at 37 °C with 5 % CO₂ in DMEM/BSA. Images were acquired for 120 min at one frame/20 s in the presence of PI using a spinning disk fluorescence microscope (UltraVIEW VoX, PerkinElmer with a 40 × 1.3 N.A. oil objective). Shown are representative videos of XY (top) and XZ (bottom) views. Time is displayed as minutes: seconds after the cell contacted the mHEL-GFP expressing COS cell. The video is displayed at 15 frames/s. Bar, 5 µm.

https://elifesciences.org/articles/66984/figures#video7

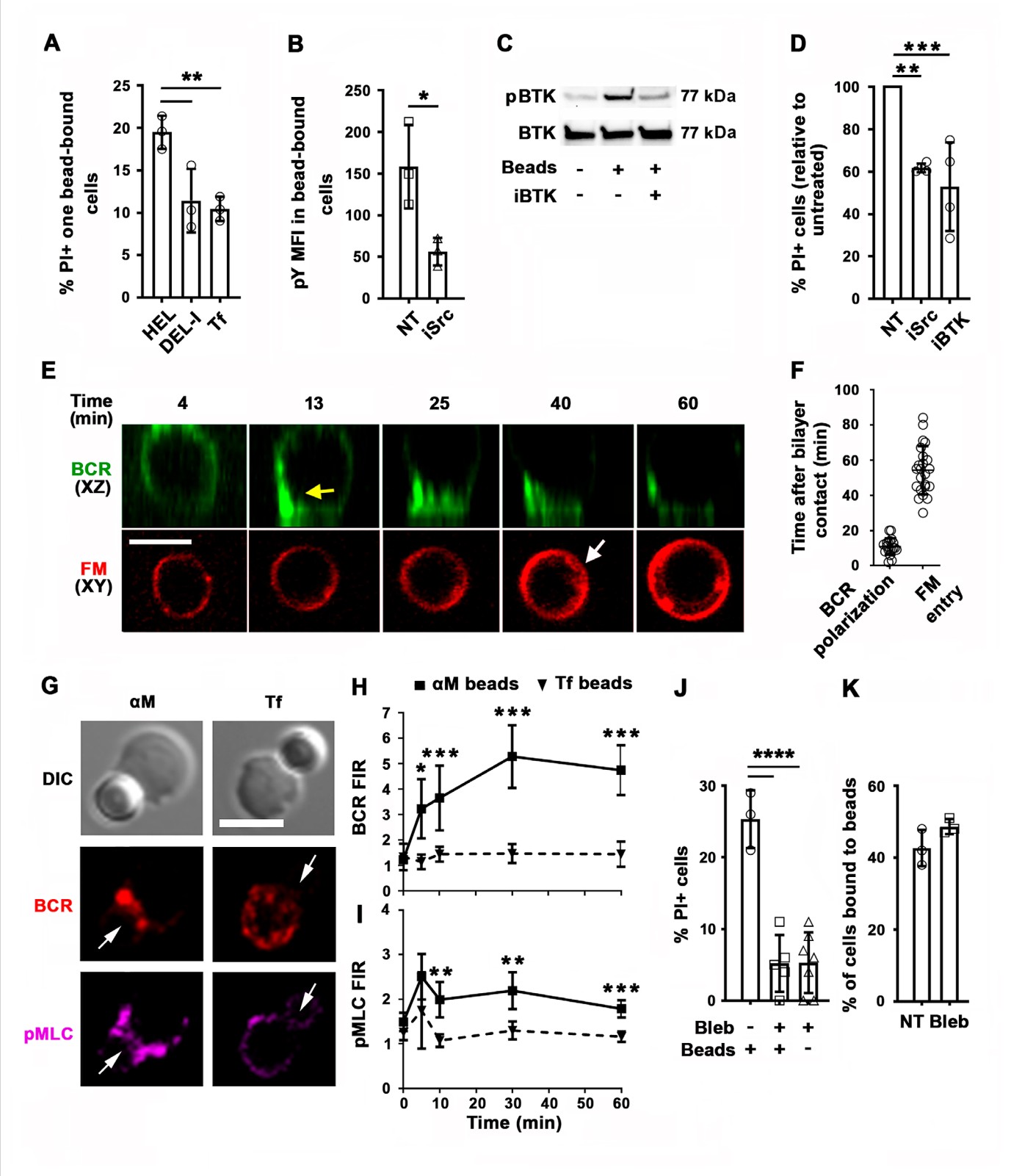

**Figure 4.** PM permeabilization induced by surface-associated antigen depends on high-affinity BCR-antigen binding, BCR signaling, and non-muscle myosin II (NMII) motor activity. (**A**) Percentages of PI+ single bead-binding B-cells after incubation with HEL-, DEL-I- or Tf-beads (1:4 cell:bead ratio) for 30 min. Data points represent independent experiments (mean ± SD). (**B**) Mean fluorescence intensity (MFI) of phosphotyrosine (pY) in HEL-bead-bound B-cells treated or untreated (NT) with a Src kinase inhibitor (iSrc) by flow cytometry. Data points represent independent experiments (mean ±

*Figure 4 continued*

SD). (**C**) Western blot analysis of phosphorylated BTK (pBTK) and BTK in B-cells incubated with HEL-beads in the presence or absence of a BTK inhibitor (iBTK) for 30 min. (**D**) Percentages of PI+ HEL-bead-bound cells treated with iSrc or iBTK relative to not-treated (NT) at 30 min. Data points represent independent experiments (mean ± SD). (**E**) Spinning disk time-lapse images of BCR polarization (yellow arrow) in a B-cell incubated with αM-PLB in the presence of FM4-64 (white arrow, intracellular FM). (**F**) Timing of BCR polarization and FM entry of individual cells interacting with αM-PLB (*Video 8*). Data points represent individual cells in three independent experiments (mean ± SD). (**G**) Confocal images of BCR and phosphorylated NMII light chain (pMLC) staining in B-cells interacting with αM- or Tf-beads (arrows, bead binding sites). (**H and I**) FI ratio (FIR) of BCR (**H**) and pMLC (**I**) staining at the bead-binding site relative to the opposite PM in αM- and Tf-bead-bound cells over time. Data represent the averages of three independent experiments (mean ± SD). (**J**) Percentages of PI+ bead-binding B-cells incubated with αM-beads for 30 min with or without blebbistatin (Bleb). Data points represent individual videos from three independent experiments (mean ± SD). (**K**) Percentages of bead-bound B-cells incubated with αM-beads for 30 min in the presence or absence of Bleb. Data points represent independent experiments (mean ± SD). Bars, 5 µm. *p ≤ 0.05, **p ≤ 0.01, ***p ≤ 0.005, ****p ≤ 0.001, unpaired Student's *t*-test (**B, H, I, K**) or one-way ANOVA (**A, D, J**).

The online version of this article includes the following figure supplement(s) for figure 4:

**Figure supplement 1.** Impact of BCR-antigen affinity on B-cell-bead binding.

**Figure supplement 2.** B-cell binding to αM-PLB but not to Tf-PLB triggers BCR polarization first and PM permeabilization later.

**Figure supplement 3.** BCR and phosphorylated myosin light chain (pMLC) polarize toward αM-bead binding sites.

required for the resealing of PM wounds (*Reddy et al., 2001*). To determine if permeabilization by surface-associated αM or HEL triggered exocytosis of lysosomes in B-cells, we first examined whether luminal epitopes of the lysosomal membrane protein LIMP-2 were exposed on the cell surface. Flow cytometry detected surface LIMP-2 in a higher percentage of B-cells binding αM-beads than in B-cells binding Tf-beads (*Figure 5A and B*). Notably, surface exposure of LIMP-2 was lower in MD4 B-cells binding DEL-I-beads compared to HEL-beads (*Figure 5C*). These results reveal a close correlation between the extent of PM permeabilization (*Figure 4A*) and lysosomal exocytosis

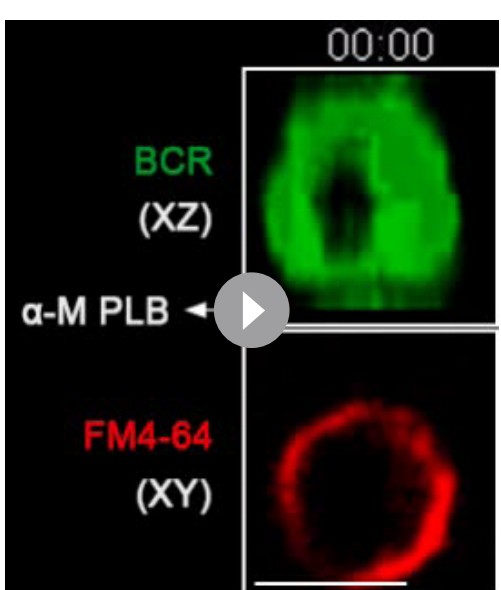

**Video 8.** The BCR polarizes towards antigen-binding sites before PM permeabilization. Splenic B-cells stained with anti-BCR antibodies were added to αM-PLB and imaged in a live imaging chamber at 37 °C with 5 % $CO_2$ in DMEM/BSA. Images were acquired for 60 min at one frame/20 s in the presence of FM4-64 using a spinning disk fluorescence microscope (UltraVIEW VoX, PerkinElmer with a 60 × 1.4 N.A. oil objective). Top: XZ view showing BCR (green) polarization towards the αM-PLB (white arrow). Bottom: XY view showing intracellular influx of FM4-64 (red, yellow arrow). Time is displayed as minutes: seconds after the cell contacted the αM-PLB. The video is displayed at 15 frames/s. Bar, 5 µm.
https://elifesciences.org/articles/66984/figures#video8

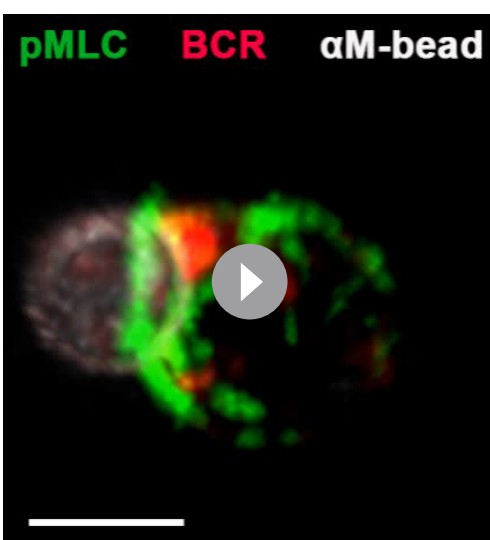

**Video 9.** BCR and phosphorylated non-muscle myosin II (pMLC) polarize towards αM-bead-binding sites on a B-cell. Shown is a 3D representation of co-polarization of the BCR (red) and pMLC (green) towards the site of αM-bead (white) binding in a splenic B-cell. Z-stack images were acquired using a Zeiss LSM710 confocal fluorescence microscope (63 × 1.4 N.A. oil objective) and the 3D reconstruction was generated using Volocity software (PerkinElmer). Bar, 3 µm.
https://elifesciences.org/articles/66984/figures#video9

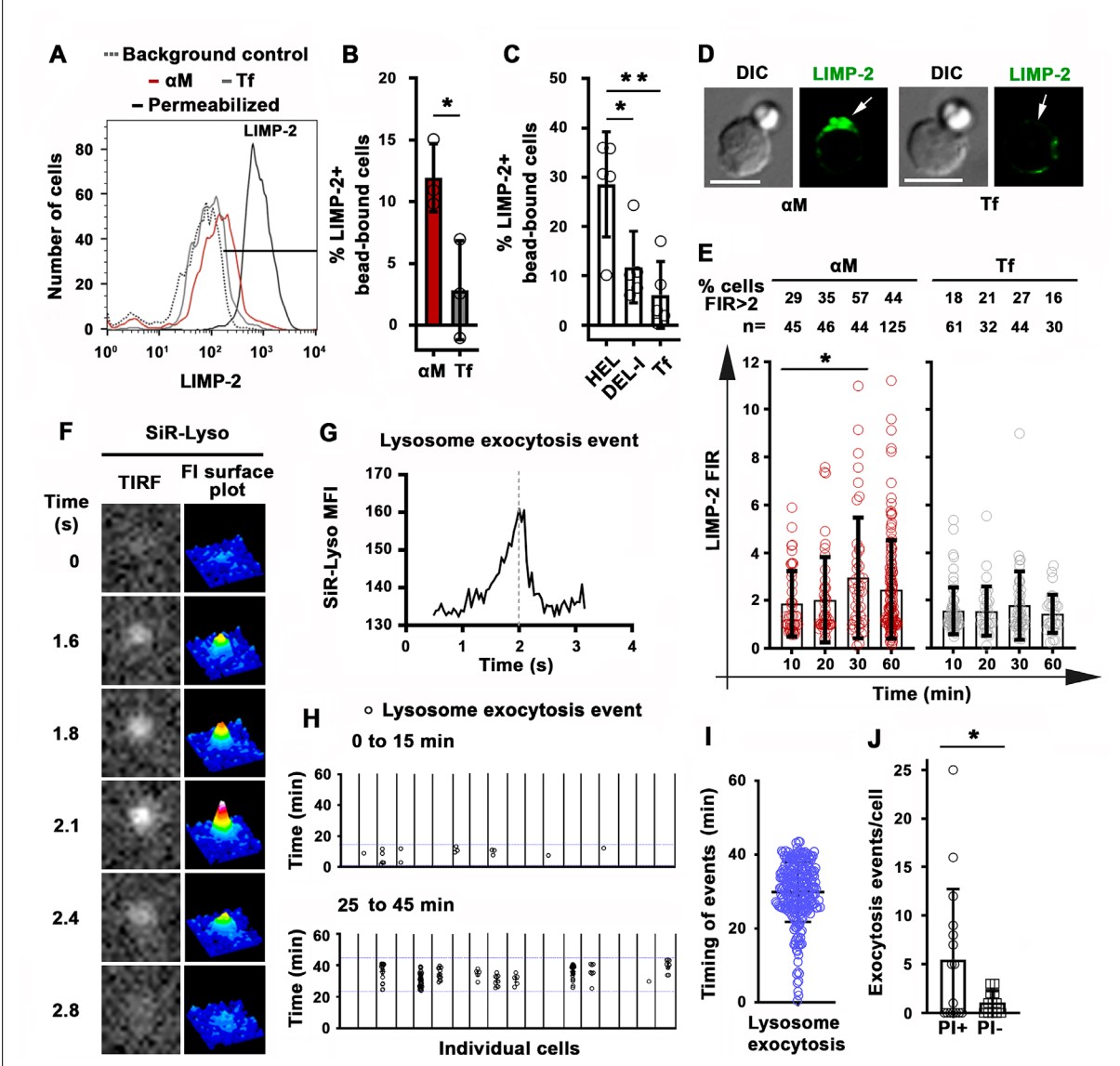

**Figure 5.** Antigen-induced permeabilization triggers lysosomal exocytosis. (**A**) Flow cytometry analysis of surface-exposed (no detergent permeabilization) and/or intracellular LIMP-2 (with detergent permeabilization) of bead-bound B-cells after incubation with αM- or Tf-beads for 30 min, showing 3000 cells per condition. (**B and C**) Percentages of cells with surface-exposed LIMP-2 (relative to values with secondary antibody alone) in bead-bound B-cells incubated with αM- or Tf-beads (**B**) or with HEL-, DEL-I- or Tf-beads (**C**) for 30 min. Data points represent independent experiments (mean ± SD). (**D**) Confocal images of surface-exposed LIMP-2 in B-cells incubated with αM- or Tf-beads (arrows, bead-binding sites). (**E**) FIR (bead-binding site:opposite PM) of surface-exposed LIMP-2 in individual cells over time. Data points represent individual cells (mean ± SD). (**F**) Total internal reflection microscopy (TIRF) images (left) and FI surface plots (right) of SiR-Lyso at the B-cell surface contacting αM-PLB (*Video 10*). (**G**) Representative MFI versus time plot of a SiR-Lyso-loaded lysosome undergoing exocytosis. (**H**) SiR-Lyso exocytosis events (circles) in individual B-cells during the first 0–15 min or 25–45 min of incubation with αM-PLB. (**I**) Timing of individual SiR-Lyso exocytosis events in B-cells incubated with αM-PLB for 45 min. Data points represent individual SiR-Lyso exocytosis events from three independent experiments (mean ± SD). (**J**) Numbers of SiR-Lyso exocytosis events per B-cell permeabilized (PI+) or not permeabilized (PI-) by αM-PLB during 45 min. Data points represent individual cells from three independent experiments (mean ± SD). *p ≤ 0.05, **p ≤ 0.01, unpaired Student's *t*-test (**B and J**) or one-way ANOVA (**C and E**). Bars, 5 μm.

The online version of this article includes the following figure supplement(s) for figure 5:

**Figure supplement 1.** BCR-mediated binding of αM-beads induces surface exposure of the LIMP-2 luminal domain at bead contact sites.

**Figure supplement 2.** Detection of lysosomal exocytosis by TIRF microscopy.

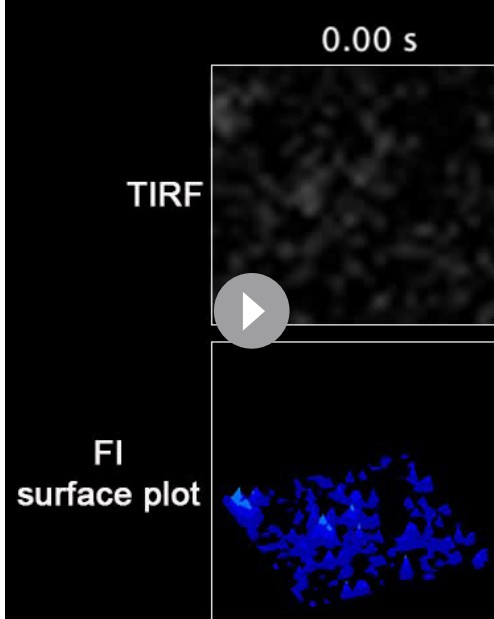

**Video 10.** A lysosomal exocytosis event detected by total internal reflection fluorescence (TIRF) microscopy. Splenic B-cells preloaded with SiR-Lyso were incubated with α M-PLB in a coverslip chamber at 37 °C with 5 % $CO_2$ in DMEM/BSA for 30 min. Time-lapse images were acquired for 20 min at eight frames/s using a TIRF microscope (NIKON Eclipse Ti-E TIRF, 63 × 1.49 NA oil objective). Top: TIRF images of a lysosome appearing in the TIRF evanescent field and then rapidly losing the SiR-Lyso signal due to fusion with the B-cell PM. Bottom: FI surface plot corresponding to the video on the top. Time is displayed in seconds. The video is displayed at 15 frames/s.

https://elifesciences.org/articles/66984/figures#video10

induced by surface-associated αM, HEL or DEL-I (*Figure 5B–C*). Surface LIMP-2 was predominantly detected at sites of αM-bead binding (*Figure 5D* and *Figure 5—figure supplement 1*) and this polarized pattern, measured by FIR, increased after ~30 min of interaction with αM- but not Tf-beads (*Figure 5E*). Notably, this time-frame was similar to the average period required for PM permeabilization (*Figure 2E*). Next, we performed live total internal reflection fluorescence (TIRF) microscopy of B-cells preloaded with the luminal lysosomal probe SiR-Lyso (a membrane-permeable fluorescent peptide that binds to the lysosomal enzyme cathepsin D) while contacting αM-PLB. Exocytosis events were identified by rises in the fluorescence intensity of SiR-Lyso puncta (reflecting lysosome entry into the TIRF evanescent field adjacent to the PM) followed by sharp decreases within ~2 s (reflecting dye dispersion upon fusion of lysosomes with the PM) (*Figure 5F and G*, *Figure 5—figure supplement 2* and *Video 10*). Exocytosis events were observed in the majority of individual PI-positive cells interacting with αM-PLB (*Figure 5H*) and occurred predominantly ~30–45 min after αM-PLB contact (*Figure 5H and I*), a timing similar to PM permeabilization and LIMP-2 exposure. Lysosomal exocytosis events were significantly more frequent in permeabilized B-cells when compared to B-cells that remained intact (*Figure 5J*). These results show that permeabilization of B-cells by surface-associated antigen triggers exocytosis of lysosomes.

We next determined if B-cells were capable of resealing their PM, by using an assay involving sequential exposure to two different membrane-impermeable fluorescent dyes (*Reddy et al., 2001*). Resealed cells were quantified by flow cytometry as the percentage of permeabilized cells binding αM-beads (stained intracellularly with FM4-64 kept throughout the assay) that excluded the membrane-impermeable dye SYTOX Blue (added only during the last 10 min of the assay) (*Figure 6A*). Under these conditions, ~50 % of B-cells permeabilized by surface-associated antigen resealed their PM within the assay period (*Figure 6B*). Inhibition of lysosomal exocytosis with bromoenol lactone (BEL) (*Fensome-Green et al., 2007*; *Tam et al., 2010*) significantly reduced the percentage of resealed cells (*Figure 6A and B*). We found no evidence that the reduction in resealed cells after BEL treatment was due to toxicity of this inhibitor. B-cell populations with low forward-scatter versus side-scatter values typical of dead cells did not increase after BEL treatment (*Figure 6—figure supplement 1*). Exposure to BEL also did not increase the small fraction ( < 7%) of Tf-bead-binding B-cells that was permeable to SYTOX Blue (*Figure 6—figure supplement 1*). These data suggest that lysosomal exocytosis is required for the resealing of B-cells permeabilized by binding to surface-associated antigen. To confirm that individual antigen-permeabilized B-cells resealed, we used live imaging to visualize cells incubated with αM-PLB in the presence of SYTOX Green. PI was then added for the last 10 min of the 4 hr incubation. Time-lapse images showed that B-cells that became permeable to SYTOX Green during interaction with αM-PLB subsequently excluded PI – a direct indication that their PM resealed during the 4 hr assay period (*Figure 6C* and *Video 11*). As expected, cells that were already permeable to SYTOX Green at the beginning of the incubation (likely non-viable cells that were damaged prior to the incubation) were

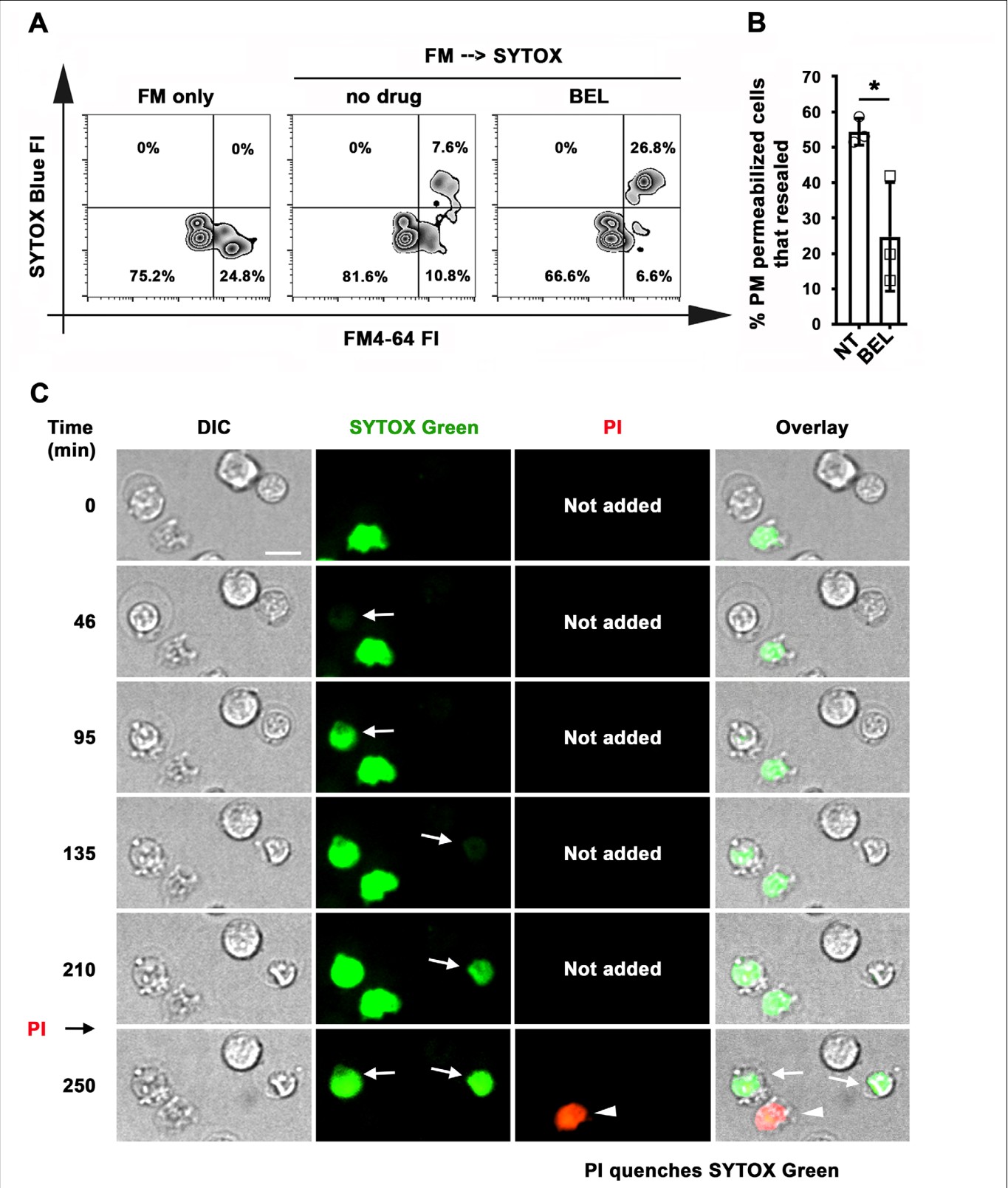

**Figure 6.** Antigen-permeabilized B-cells reseal their PM in a lysosomal exocytosis-dependent manner. (**A**) B-cells were incubated with αM-beads and permeabilized/resealed cells were assessed by flow cytometry of FM4-64 (added from the start) and SYTOX Blue (added in the last 10 min) FI, in the presence or absence of BEL. (**B**) Percentages of permeabilized αM-bead-bound cells that resealed in the presence or absence of BEL. Data points represent independent experiments (mean ± SD). (**C**) Time-lapse images of splenic B-cells incubated with αM-PLB in the presence of SYTOX Green. PI

*Figure 6 continued on next page*

*Figure 6 continued*

was added for 10 min at the end (*Video 11*). Arrows, cells that became permeabilized after contacting the αM-PLB and later excluded PI; arrowhead, cell that was SYTOX + since the start of the video and did not exclude PI. *p ≤ 0.05, unpaired Student's *t*-test (**B**). Bar, 5 μm.

The online version of this article includes the following figure supplement(s) for figure 6:

**Figure supplement 1.** BEL does not affect the PM integrity and viability of B-cells.

**Figure supplement 2.** B-cell morphological changes occurring during permeabilization by surface-associated antigen are reversible.

also permeable to PI (which causes strong quenching of the SYTOX green fluorescence upon entering cells - *Figure 6C* and *Video 11*). Interestingly, primary B-cells permeabilized during interaction with αM-beads (*Figure 1A* and *Video 1*) or αM-PLB (*Figures 1H and 6C* and *Video 11*) often displayed a shape change visualized as an increase in cell diameter, but after resealing this morphological change was gradually reversed (*Figure 6C*, *Figure 6—figure supplement 2*, *Videos 11 and 12*). Collectively, our results indicate that B-cell PM permeabilization by binding to surface-associated antigen is a reversible event, and that lysosomal exocytosis is required for PM resealing as previously shown for other cell types (*Andrews et al., 2014*).

## B-cell permeabilization and lysosomal exocytosis facilitate internalization and presentation of surface-associated antigen

We investigated the relationship between PM permeabilization by surface-associated antigen and antigen internalization using fluorescent αM covalently bound to beads or tethered to PLB. Live imaging detected αM puncta moving away from bead-binding sites into B-cells, increasing progressively between 30 and 60 min of interaction (*Figure 7A and B* and *Video 13*). In contrast, intracellular fluorescent puncta were markedly less abundant during the same time period in cells not binding αM-beads, or binding Tf-beads (*Figure 7B*). Inhibition of antigen-mediated PM permeabilization with blebbistatin significantly reduced extraction and internalization of αM coupled to beads (*Figure 7C*). When similar experiments were performed with PLB, the fraction of cells containing internalized αM and the total amount of αM uptake were significantly higher in permeabilized cells with high levels of intracellular FM staining (FM-high), compared to non-permeabilized cells with low FM staining (FM-low) (*Figure 7D–F*). These data suggest that αM-induced PM permeabilization, rapidly followed by lysosomal exocytosis, promotes extraction and internalization of αM from non-internalizable surfaces.

Next, we investigated whether antigen internalization enhanced by PM permeabilization and lysosomal exocytosis impacts antigen presentation by B-cells. We compared levels of IL-2

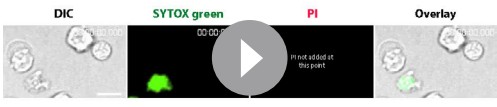

**Video 11.** B-cells exclude a second membrane-impermeable tracer after antigen-dependent permeabilization. Splenic B-cells were added to αM-PLB and imaged in a live imaging chamber at 37 °C with 5 % $CO_2$ in DMEM 2 % of FBS in the presence of SYTOX Green (green). Images were acquired for 4 hr at one frame/30 s using a spinning disk fluorescence microscope (UltraVIEW VoX, PerkinElmer with a 60 × 1.4 N.A. oil objective). PI (red) was added for 10 min at the end of the time-lapse image acquisition. The video is displayed as minutes: seconds after the cell contacted the αM-PLB. White arrows indicate cells that became permeabilized and later excluded PI. The yellow arrow indicates a cell that was stained by SYTOX Green since the beginning of the video and was not able to exclude PI. The video is displayed at 20 frames/s. Bar, 5 μm.

https://elifesciences.org/articles/66984/figures#video11

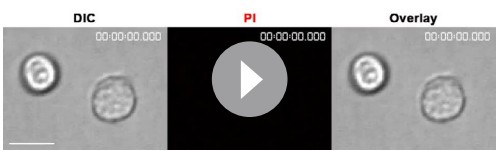

**Video 12.** B-cell morphological changes occurring during permeabilization by surface-associated antigen are reversible. Splenic B-cells were added to αM-PLB and imaged in a live imaging chamber at 37 °C with 5 % $CO_2$ in DMEM without phenol red containing 2 % FBS in the presence of PI (red). Images were acquired for 4 hr at one frame/30 s using a spinning disk fluorescence microscope (UltraVIEW VoX, PerkinElmer with a 60 × 1.4 N.A. oil objective). Time is displayed as minutes: seconds after the cells first contacted the αM-PLB. The arrow points to a cell that became permeabilized. The dashed line indicates the maximum diameter of the B-cell after permeabilization. The video is displayed at 20 frames/s. Bar, 5 μm.

https://elifesciences.org/articles/66984/figures#video12

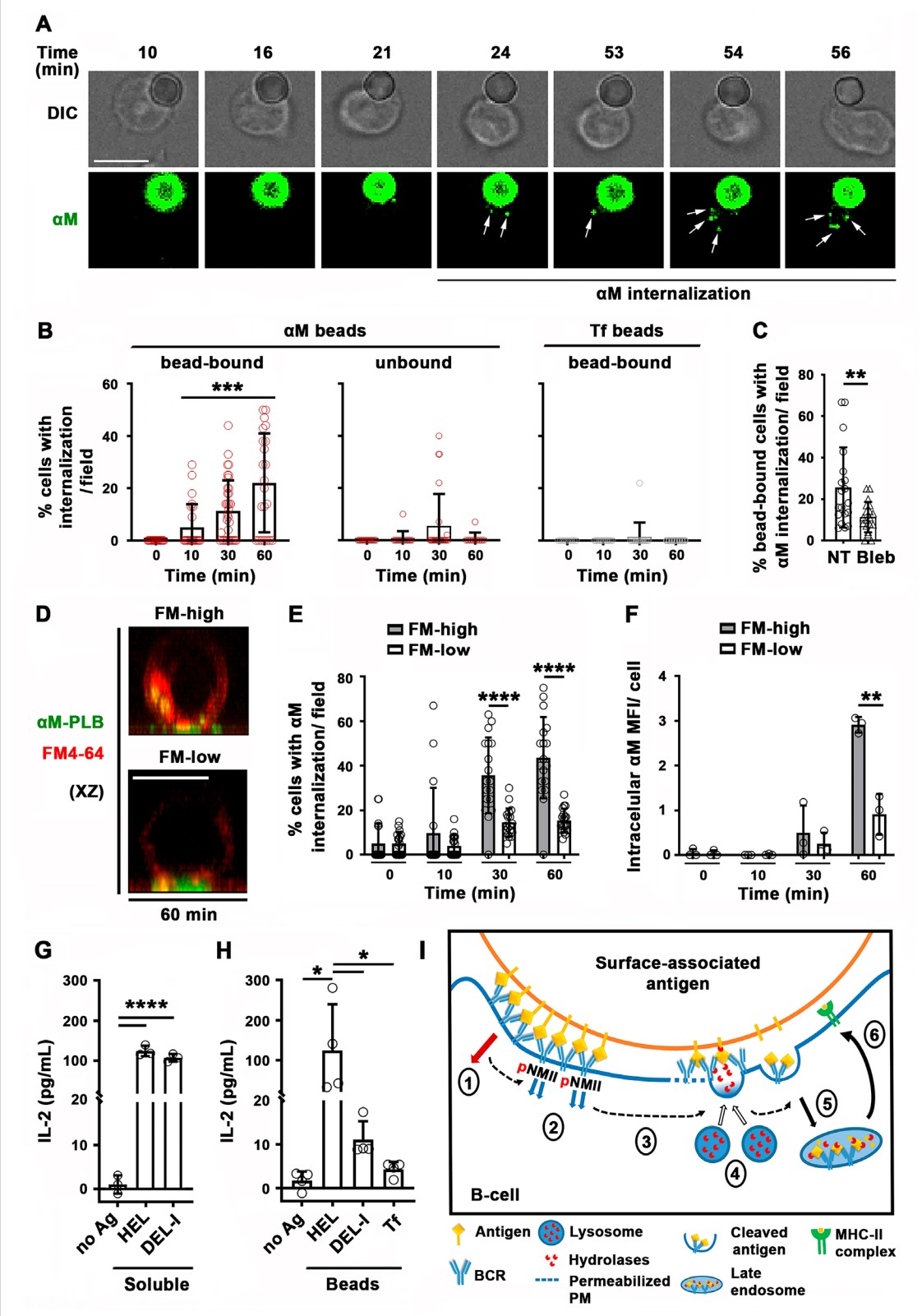

**Figure 7.** Antigen-induced PM permeabilization promotes antigen internalization and presentation. (**A**) Confocal live imaging of a B-cell interacting with fluorescent αM-beads (arrows, internalized αM). (**B**) Percentages of cells containing internalized αM or Tf, bound or not to αM- or Tf-beads, over time. Data points represent individual fields in three independent experiments (mean ± SD). (**C**) Percentages of bead-bound B-cells with internalized αM in the presence or absence of Bleb after 60 min. Data points represent individual fields in four independent experiments (mean ± SD). (**D**) Confocal

*Figure 7 continued on next page*

*Figure 7 continued*

images (xz) of αM internalization in B-cells permeabilized (FM-high) or not permeabilized (FM-low) by αM-PLB after 60 min. (**E**) Percentages of B-cells, permeabilized (FM-high) or not permeabilized (FM-low) by αM-PLB, containing internalized αM over time. Data points represent individual fields in three independent experiments (mean ± SD). (**F**) MFI values of internalized αM in individual B-cells permeabilized (FM-high) or not (FM-low) by αM-PLB over time. Data points represent independent experiments (mean ± SD). (**G**) IL-2 secretion by 3A9 T-cells activated by B-cells incubated with or without (no Ag) soluble HEL or DEL-I (10 μg/ml) for 72 hr. Data points represent independent experiments (mean ± SD). (**H**) IL-2 secretion by 3A9 T-cells activated by B-cells incubated with or without HEL-, DEL-I- or Tf-beads (1:4 cell:bead ratio) for 72 hr. Bars, 5 μm. Data points represent independent experiments (mean ± SD). *p ≤ 0.05, **p ≤ 0.01, **p ≤ 0.005, ****p ≤ 0.0001, unpaired Student's *t*-test (**C, E, F**), one-way ANOVA (**G and H**) or Kruskal-Wallis non-parametric test (**B**). (**I**) Cartoon depicting a working model for the spatiotemporal relationship of events initiated by the interaction of the BCR with surface-associated antigen. High-affinity binding stabilizes BCR-antigen interaction and induces strong BCR signaling (1) and NMII activation (2). Activated NMII generates local traction forces that permeabilize the PM (3), triggering a localized PM repair response mediated by lysosomal exocytosis. Lysosome exocytosis releases hydrolases that cleave antigen off surfaces (4), facilitating endocytosis (5) and presentation to T-cells (6).

secretion by the 3A9 T-cell hybridoma line (*Allen and Unanue, 1984*) after activation by B-cells exposed to HEL- or DEL-I-beads. B-cells exposed to high concentrations of soluble HEL or DEL-I induced similar levels of IL-2 secretion (*Figure 7G*), demonstrating that the primary B-cells used in these assays could process and present the conserved peptide present in both HEL and DEL-I for T-cell activation. In contrast, when the B-cells were exposed to lower amounts of surface-associated antigens, B-cells exposed to HEL-beads activated T-cells to produce IL-2 at markedly higher levels than cells exposed to DEL-I-beads (*Figure 7H*). These results indicate that B-cell permeabilization resulting from high-affinity antigen-BCR interaction, with its corresponding lysosomal exocytosis response, facilitates the presentation of antigen associated with non-internalizable surfaces.

## Discussion

Extracellular release of lysosomal enzymes by B-cells was previously proposed to cleave antigens tightly associated with non-internalizable surfaces, facilitating internalization and presentation to T-cells (*Yuseff et al., 2011*; *Spillane and Tolar, 2017*). However, it was unclear which mechanism was responsible for inducing lysosomal enzyme release when B-cells engaged insoluble antigen. In this study, we show that interaction of the BCR with surface-associated antigen can permeabilize the B-cell PM, triggering lysosomal exocytosis as part of the PM repair response (*Rodríguez et al., 1997*; *Reddy et al., 2001*). Antigen-dependent PM permeabilization occurs at antigen-binding sites and is reversible under conditions that allow lysosomal exocytosis. We further demonstrate that PM permeabilization and lysosomal exocytosis require high-affinity binding of the BCR to antigen, BCR signaling and activation of NMII motor activity, and that this process facilitates antigen internalization, processing, and presentation. Thus, our study identifies a critical novel step in the affinity-dependent process by which B-cells capture antigen tightly associated with surfaces, for effective internalization and subsequent presentation to T-cells.

Capture and internalization of antigen tightly associated with surfaces is an important immunological process, as B-cells encounter this type of antigen in vivo on parasites, bacteria and viruses, as well as immune cells such as follicular dendritic cells. Follicular dendritic cells capture antigen drained into lymph nodes and present it on their surface to germinal center B-cells. In this manner, follicular dendritic cells enhance BCR antigenic stimulation by increasing antigen avidity, in addition to providing costimulatory molecules (*Natarajan et al., 2001*). While the exact percentage is unknown, studies have

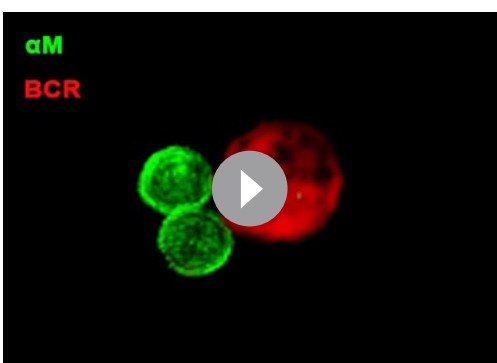

**Video 13.** B-cell with polarized surface BCRs and containing fluorescent $\alpha$ M extracted from beads. The surface BCRs of splenic B-cells were labeled with Cy3-Fab-donkey anti-mouse IgM+ G at 4 °C. Labeled B-cells were incubated with AF488-$\alpha$ M-beads at 37 °C with 5 % $CO_2$ for 60 min and then fixed. Images were acquired using a Zeiss LSM710 (63 × 1.4 N.A. oil objective), and the 3D reconstruction was generated with Volocity software (PerkinElmer). The arrow points to internalized AF488-$\alpha$ M.
https://elifesciences.org/articles/66984/figures#video13

suggested that the majority of antigens that B-cells encounter in vivo are in a membrane-associated form (*Batista and Harwood, 2009*). Importantly, the capture, internalization, and presentation of such surface-associated antigens to T-cells play a critical role in selecting specific B-cells for survival, clonal expansion and differentiation into long-lived high-affinity memory B-cells and antibody-secreting cells (*Gitlin et al., 2014*).

We found that B-cell PM permeabilization induced by surface-associated antigen depends on the motor activity of NMII. Following BCR polarization, activated NMII accumulates at sites of B-cell binding to αM- or HEL-beads or PLB before permeabilization occurs. These findings are consistent with previous studies showing that internalization of surface-associated but not soluble antigen requires NMII-mediated traction forces at antigen-binding sites (*Natkanski et al., 2013*; *Spillane and Tolar, 2018*). Collectively, our results support the notion that NMII-mediated traction forces generated during BCR-antigen interaction are responsible for permeabilization of the B-cell PM. Whether this permeabilization is due to tearing of the lipid bilayer (*Andrews et al., 2014*) or the opening of mechanosensitive membrane channels (*Liu et al., 2018*; *Liu and Ganguly, 2019*) is currently unknown. However, our finding that three distinct membrane-impermeable probes, PI, FM lipophilic dyes, and Ponceau 4 R readily gain access to the cytosol after B-cell interaction with surface-associated antigen suggest that NMII-mediated membrane tearing is the mechanism underlying antigen-dependent B-cell PM permeabilization. In this context, it is noteworthy that Endophilin A2, a protein that facilitates the resealing of PM wounds (*Corrotte et al., 2020*), also contributes to BCR-mediated internalization of membrane-associated antigen (*Malinova et al., 2021*).

We were initially surprised to observe B-cell PM permeabilization during BCR-mediated binding of surface-associated antigen, a process that is known to generate myosin-mediated forces as a mechanism to capture antigen. To confirm that permeabilization occurs, we utilized three different membrane-impermeable probes, two types of BCR ligands, and three types of presenting surfaces. All generated similar results. We first detected B-cell permeabilization during interaction with surface-associated antigen by following the entry of membrane-impermeable DNA-binding or lipophilic dyes. While these compounds bind to different intracellular structures, both showed sudden rather than gradual increases in intracellular staining, consistent with PM permeabilization. To strengthen these results, we designed an independent assay based on the ability of Ponceau 4 R to enter B-cells and quench the fluorescence of CFSE, a widely used vital dye that covalently labels cytosolic molecules without affecting cell viability. Ponceau 4 R has been used to reduce the extracellular background of fluorescence-based assays because it is membrane-impermeable and potently quenches the emission of fluorophores in the 490–560 nm range (*Tay et al., 2019*). We found that Ponceau 4 R influx rapidly quenches the fluorescence of CFSE-labeled B-cells, providing us with an independent and accurate tool to determine the kinetics of antigen-induced PM permeabilization.

We also showed that endocytosis does not account for the sudden, massive influx of lipophilic dyes that occurs in B-cells binding surface-associated antigen. Cross-linking surface BCRs with soluble antibodies (*Song et al., 1995*; *Cousin et al., 2018*), which did not permeabilize the B-cell PM, induced the endocytosis of lipophilic dyes – as expected from a tracer that is associated with the outer leaflet of the PM. However, the endocytosed lipophilic dye appeared as small puncta that gradually accumulated at the cell periphery, in sharp contrast to the sudden, massive dye influx that reaches the nuclear envelope in antigen-permeabilized cells. Consistent with this result, endocytosed fluorescent Fab' covalently attached to beads also appeared as small puncta in our live imaging assays. Thus, we conclude that the sudden, massive influx of lipophilic dyes is the result of PM permeabilization but not of dye endocytosis.

The PM of primary B-cells can be damaged by phototoxicity during prolonged live imaging, or by necrosis or apoptosis. To control for such events, in parallel to our assays with surface-associated antigen, we measured the permeabilization levels of cells interacting with Tf-beads or Tf-PLB, which bind the Tf receptor with similar affinity as antigen-BCR but without BCR activation (*Fuchs and Gessner, 2002*). Low levels of non-specific permeabilization of B-cells were detected in these controls, not surprisingly given that primary splenic B-cells are often injured during the purification process. Furthermore, we did not observe an increase in apoptotic markers in B-cells interacting with surface-associated antigen. Importantly, our assays involving sequential exposure to membrane-impermeable dyes revealed that a significant fraction of the antigen-permeabilized B-cells subsequently resealed.

Thus, our findings cannot be explained by a loss in B-cell viability, strongly suggesting that B-cells can become transiently permeabilized when binding antigen that is tightly associated with surfaces.

We found that two different model antigens, αM and HEL, can induce B-cell PM permeabilization when attached to surfaces. This shows that BCR binding through bona fide antigen-binding sites is not a requirement for generation of the mechanical forces leading to B-cell PM permeabilization. Since stiffness of the antigen-presenting surface appears to impact BCR signaling and antigen capture (*Spillane and Tolar, 2017*; *Wang et al., 2018b*), it could be argued that antigen tethered to latex beads or PLB assembled on glass coverslips represent unnaturally stiff surfaces that might cause B-cell permeabilization. To investigate this issue, we utilized COS-7 cells expressing mHEL, a surface-associated antigen previously shown to engage MD4 B-cells in vivo when expressed in mouse models (*Hartley et al., 1991*). Our finding that BCR engagement of mHEL on the surface of COS-7 cells also induces PM permeabilization supports the notion that this process occurs under physiological conditions and is likely to be relevant in vivo.

Not all B-cells binding surface-associated antigen were permeabilized, possibly due to heterogeneity of the primary B-cell population used in our assays. Splenic B-cells are found at different stages of peripheral maturation and differentiation (*Sagaert and De Wolf-Peeters, 2003*; *Allman and Pillai, 2008*), binding antigen with variable affinities at different times and generating distinct responses. In subsequent studies, it will be interesting to determine which subsets of B-cells are more effective in capturing and presenting surface-associated antigen through NMII-dependent PM permeabilization.

The rapid exocytosis of lysosomes triggered by B-cell permeabilization uncovered in our study provides a mechanistic explanation for the previously reported affinity-dependent extraction and presentation of antigen associated with non-internalizable surfaces (*Batista and Neuberger, 2000*). We showed that the low-affinity DEL-I antigen induces markedly lower levels of PM permeabilization, lysosome exocytosis, and antigen presentation when compared to the higher affinity HEL, when the two antigens are displayed on surfaces at similar densities. Surface association significantly enhances the avidity of antigens by increasing their valency, a process that can reduce the impact of BCR-binding affinity on BCR signaling, antigen internalization, and presentation when compared to soluble forms of the same antigen. However, this avidity effect is primarily observed with antigen associated with surfaces that B-cells are able to internalize (*Batista and Neuberger, 2000*), and it is known that B-cell subsets such as native follicular B-cells have very low phagocytic capacity (*Vidard et al., 1996*). We envision that when antigen is strongly associated with non-internalizable surfaces, low-affinity BCR-antigen interactions are disrupted before B-cells can extract antigen. In this scenario, high-affinity BCR interactions would be critical for sustaining antigen binding under NMII-mediated traction forces, to promote PM permeabilization, lysosomal enzyme release, and antigen extraction. High-affinity BCR-antigen binding is also expected to induce more robust signaling than low-affinity binding, enabling higher levels of NMII activation (*Fleire et al., 2006*; *Natkanski et al., 2013*) and polarization to drive PM permeabilization. Collectively, in addition to supporting the notion that tight antigen attachment to non-internalizable surfaces facilitates B-cell affinity discrimination, our results expand the mechanistic understanding of why different physical and chemical forms of immunogens impact the efficacy of vaccines (*Bachmann and Jennings, 2010*; *Khan et al., 2015*).

Lysosomal exocytosis is acutely dependent on rapid elevations in $[Ca^{2+}]_i$ (*Reddy et al., 2001*; *Jaiswal et al., 2002*). PM tears cause immediate $Ca^{2+}$ influx and massive lysosomal exocytosis (*Reddy et al., 2001*; *Tam et al., 2010*), due to the markedly higher $Ca^{2+}$ concentration in the extracellular space compared to the cytoplasm. BCR engagement of antigen also induces $[Ca^{2+}]_i$ increases (*Baba and Kurosaki, 2016*; *Tanaka and Baba, 2020*), and we cannot rule out the possibility that BCR-mediated $Ca^{2+}$ fluxes might contribute to the initiation of lysosomal exocytosis, which would then be amplified by PM permeabilization and more robust $Ca^{2+}$ influx. However, while BCR-induced $Ca^{2+}$ fluxes occur in most antigen-binding B-cells within seconds of antigen binding, we found that the majority of lysosomal exocytosis and antigen internalization events occur >30 min after antigen binding, a time frame that coincides with the period required for antigen-induced PM permeabilization. Thus, our data suggest that BCR-mediated $[Ca^{2+}]_i$ increases are unlikely to be the primary driver of the lysosomal exocytosis events that facilitate endocytosis of surface-associated antigen. However, BCR-triggered $Ca^{2+}$ fluxes may have induced the small number of initial lysosomal exocytosis events that we detected during the first 15 min of B-cell interaction with surface-associated antigen. It is also conceivable that

early BCR-induced Ca$^{2+}$ fluxes contribute to antigen-induced B-cell PM permeabilization by activating NMII and actin reorganization (*Izadi et al., 2018*).

Collectively, our results provide important insights into the spatiotemporal relationship of events initiated by interaction of the BCR with surface-associated antigen (*Figure 7I*). Our findings suggest that high-affinity binding stabilizes BCR-antigen interactions, inducing strong BCR signaling and NMII activation to locally generate traction forces that permeabilize the PM. Ca$^{2+}$ entry would then trigger a localized PM repair response mediated by lysosomal exocytosis, releasing hydrolases that can cleave antigen off surfaces, facilitating endocytosis and presentation to T-cells. Our results support the notion that B-cells utilize a cellular mechanism that evolved for surviving PM injury to promote the acquisition, presentation, and possibly affinity discrimination of surface-associated antigens.

# Materials and methods

## Key resources table

| Reagent type (species) or resource | Designation | Source or reference | Identifiers | Additional information |
|---|---|---|---|---|
| Cell line (*Mus musculus*) | A20 | ATCC | TIB-208 | B-cell lymphoma |
| Cell line (*Mus musculus*) | 3A9 | ATCC | CRL-3293 | T-cell hybridoma |
| Cell line (*Cercopithecus aethiops*) | COS-7 | ATCC | CRL-1651 | Kidney fibroblasts |
| Biological sample (*Mus musculus*) | WT (C57BL/6) | Jackson Laboratories | 000664 | Primary B-cells freshly isolated from C57BL/6' spleen |
| Biological sample (*Mus musculus*) | MD4 (C57BL/6-Tg (IghelMD4)4Ccg/J) | Jackson Laboratories | 002595 | Primary B-cells freshly isolated from C57BL/6-Tg (IghelMD4)4Ccg/J's spleen |
| Biological sample (*Mus musculus*) | B10.BR-H2$^{K2}$ H2-T18$^a$/ SgSnJJrep | Jackson Laboratories | 004804 | Primary B-cells freshly isolated from B10.BR-H2$^{K2}$ H2-T18$^a$/ SgSnJJrep's spleen |
| Antibody | Anti-phosphotyrosine mAb 4G10 (mouse monoclonal) | Millipore | 05–321 | 1:500 |
| Antibody | AF488-anti-mouse IgG$_{2b}$ (goat polyclonal) | Thermo Fisher Scientific | A-21141 | 1:500 |
| Antibody | AF647-anti-goat IgG (H + L) (donkey polyclonal) | Invitrogen | A21447 | 10 µg/ml |
| Antibody | Anti-cleaved caspase-3 (Asp175) (rabbit polyclonal) | Cell Signaling | 9661T | 1:500 |
| Antibody | Cy5-Fab anti-mouse IgG (donkey polyclonal) | Jackson ImmunoResearch | 715-175-151 | 5 µg/ml |
| Antibody | AF488-anti-rabbit IgG (H + L) highly cross-adsorbed secondary antibody (donkey polyclonal) | Thermo Fisher Scientific | A-21206 | 1:200 |
| Antibody | Anti-BTK (rabbit monoclonal) | Cell Signaling | 8,547 | 1:1,000 |
| Antibody | Anti-phospho-BTK (rabbit monoclonal) | Abcam | 68,217 | 1:500 |
| Antibody | HRP-anti-rabbit (goat polyclonal) | Jackson Immune Research | 111-035-144 | 1:1,000 |
| Antibody | Cy3-Fab-anti–mouse IgM+ G (goat polyclonal) | Jackson Immune Research | 115-165-166 | 1:200 |
| Antibody | Anti-phosphorylated myosin light chain (pMLC) (rabbit polyclonal) | Cell Signaling | 3,671 S | 1:50 |
| Antibody | AF633-anti-rabbit IgG (goat polyclonal) | Invitrogen | A-21070 | 1:500 |
| Antibody | Anti-LIMP-2 (rabbit polyclonal) | Sigma-Aldrich | SAB3500449-100UG | 1:200 |

*Continued on next page*

*Continued*

| Reagent type (species) or resource | Designation | Source or reference | Identifiers | Additional information |
|---|---|---|---|---|
| Antibody | AF488 donkey-anti-rabbit IgG (donkey polyclonal) | Life technology | A32790 | 1:200 |
| Antibody | Anti-CD90.2 (rat monoclonal) | Biolegend | 105,310 | 1 µl/ $2 \times 10^6$ cells |
| Antibody | $\alpha$ M (F(ab')$_2$ goat-anti-mouse IgM+ G) (goat polyclonal) | Jackson Immune Research | 115-006-068 | Binds to BCR |
| Antibody | AF488-$\alpha$ M AffiniPure F(ab')$_2$ fragments of anti- mouse IgG (H + L) (goat polyclonal) | Jackson Immune Research | 115-546-003 | Binds to BCR |
| Antibody | Biotin-SP (long spacer)-conjugated Fab fragments of anti-mouse IgG (H + L) (goat polyclonal) | Jackson Immune Research | 115-067-003 | |
| Commercial assay or kit | SiR-Lysosome and Verapamil | Cytoskeleton | CY-SC012 | Lysosome probe 1 µM and 10 µM |
| Commercial assay or kit | IL-2 ELISA kit | Biolegend | 431,804 | |
| Commercial assay or kit | BCA kit | Thermo Fisher Scientific | 23,235 | Protein measurement during bead preparation |
| Software, algorithm | Volocity Suite | PerkinElmer | | https://ir.perkinelmer.com/news-releases/news-release details/perkinelmer-launches-volocityr-60-high-performance-3d-cellular |
| Software, algorithm | NIH Image J | NIH | | https://imagej.nih.gov/ij/ |
| Software, algorithm | MATLAB | MathWorks | | https://www.mathworks.com/products/matlab.html |
| Software, algorithm | Prism | GraphPad | | https://www.graphpad.com/scientific-software/prism/ |
| Chemical compound, drug | Staurosporine | Abcam | 120,056 | Apoptosis induction (1 µM) |
| Chemical compound, drug | PP2 | Millipore-Sigma | 529,573 | Src kinase inhibitor (5 µM) |
| Chemical compound, drug | AVL-292 | Selleckchem | S7173 | BTK inhibitor (10 nM) |
| Chemical compound, drug | BEL (Bromoenol lactone) | Sigma-Aldrich | B1552 | 12 µM |
| Chemical compound, drug | Blebbistatin | Sigma-Aldrich | B0560 | 50 µM |
| Chemical compound, drug | Latex NH$_2$-beads | Polysciences | 17145–5 | |
| Chemical compound, drug | HEL (hen egg lysozyme) | Sigma-Aldrich | L6876 | Binds to BCR from MD4 mice |
| Chemical compound, drug | DEL-1 (duck egg lysozyme) | David B. Langley and Daniel Christ laboratory | | Binds to BCR from MD4 mice |
| Chemical compound, drug | Tf (holo-transferrin) | Sigma-Aldrich | T0665-50MG | Binds to transferrin receptor |

*Continued*

| Reagent type (species) or resource | Designation | Source or reference | Identifiers | Additional information |
|---|---|---|---|---|
| Chemical compound, drug | Biotinylated transferrin (Tf-PLB) | Sigma-Aldrich | T3915-5MG | |
| Chemical compound, drug | Streptavidin-conjugated Yellow-Green latex beads | Polysciences | 24159–1 | |
| Chemical compound, drug | Propidium iodide | Sigma-Aldrich | P4170-10MG | 50 µg/ml |
| Chemical compound, drug | FM1-43FX | Thermo Fisher Scientific | F35355 | 10 µg/ml |
| Chemical compound, drug | FM4-64FX | Thermo Fisher Scientific | F34653 | 10 µg/ml |
| Chemical compound, drug | SYTOX Blue | Invitrogen | S11348 | 300 nM |
| Chemical compound, drug | SYTOX Green | Invitrogen | S7020 | 300 nM |
| Chemical compound, drug | Guinea pig complement | Innovative Research | IGGPCSER | 100 µl/ $4 \times 10^7$ cells |
| Chemical compound, drug | 1,2-dioleoyl-sn-glycero-3-phosphocholine | Avanti Polar Lipids | 850375 P | 5 mM (PLB) |
| Chemical compound, drug | 1,2-dioleoyl-sn-glycero-3-phospho ethanolamine-cap-biotin | Avanti Polar Lipids | 870273 C | 50 µM (PLB) |
| Chemical compound, drug | Ponceau 4 R | Sigma-Aldrich | 18,137 | 1 mM |
| Chemical compound, drug | CFSE | Thermo Fisher Scientific | C34553 | 1 µM |
| Chemical compound, drug | Lipofectamine 3,000 | Thermo Fisher Scientific | L3000008 | |
| Chemical compound, drug | Bovine fibronectin | Millipore | 341,631 | 5 mg/ml |
| Chemical compound, drug | AF88-Tf (transferrin from human serum, Alexa Fluor 488 conjugate) | Thermo Fisher Scientifc | T13342 | Binds to transferrin receptor |
| Transfected construct (*Cercopithecus aethiops*) | mHEL-GFP | Michael R. Gold laboratory | *Wang et al., 2018a* (DOI: 10.1007/978-1-4939-7474-0_10) | Wild-type HEL protein, the complete EGFP protein, the transmembrane region of H-2K[b], and the 23-amino acid cytoplasmic domain of H-2K[b] |

## Mice, B-cell isolation, and culture

Primary B-cells were isolated from the spleens of wild-type C57BL/6, MD4 transgenic (C57BL/6 background), B10.BR-*H2^{k2} H2-T18^{a}*/SgSnJJrep (Jackson Laboratories), and F1 of B10.BR-*H2^{k2} H2-T18^{a}*/SgSnJJrep x MD4 mice using a previously published protocol (*Miller et al., 2015*). Briefly, mononuclear cells were isolated by Ficoll density-gradient centrifugation (Sigma-Aldrich). T-cells were removed with anti-mouse CD90.2 mAb (BD Biosciences) and guinea pig complement (Innovative Research, Inc) and monocytes and dendritic cells by panning. B-cells were kept at 37 °C and 5 % $CO_2$ before and during experiments. All procedures involving mice were approved by the Institutional Animal Care and Usage Committee of the University of Maryland.

The A20 B-cell lymphoma line (ATCC #TIB-208) was cultured in DMEM (Lonza) supplemented with 10 % of FBS (Thermo Fisher Scientific), 0.05 mM 2-mercaptoethanol (Sigma-Aldrich), 10 mM MOPS, 100 units/ml penicillin, and 100 µg/ml streptomycin (Gemini) at 37 °C and 5 % $CO_2$. The 3A9 T-cell hybridoma line (ATCC #CRL-3293) was cultured in DMEM (ATCC) supplemented with 5 % FBS (Thermo

Fisher Scientific), 0.05 mM 2-mercaptoethanol (Sigma-Aldrich) at 37 °C and 5 % $CO_2$. ATCC follows the highest manufacturing standards and uses the most reliable procedures to verify and authenticate every cell line and to ensure there is no mycoplasma contamination.

## Antigen-coated beads

Latex $NH_2$-beads (3 µm diameter, $3.5 \times 10^8$ beads/preparation, Polysciences) were activated with 8 % glutaraldehyde in 0.5 ml PBS for 120 min under rotation at room temperature, washed with PBS, and incubated overnight with equal molar amounts of F(ab')₂ goat-anti-mouse IgM+ G (αM, 20 µg/ml, Jackson ImmunoResearch Laboratories), hen egg lysozyme (HEL, 5.8 µg/ml, Sigma-Aldrich), duck egg lysozyme (DEL)-I (*Langley et al., 2017*), holo-transferrin (Tf, 32 µg/ml, Sigma-Aldrich), Alexa Fluor (AF) 488-conjugated Tf (AF488-Tf, 32 µg/ml, Thermo Fisher Scientific), or AF488-F(ab')₂ goat-anti-mouse IgM+ G (AF488-αM, 20 µg/ml, Jackson ImmunoResearch Laboratories) in 1 ml PBS. Protein content determination (BCA, Thermo Fisher Scientific) of coupling solutions before and after bead incubation confirmed that similar molar amounts of protein were conjugated in each case. The beads were then blocked with PBS 1 % BSA for 30 min under rotation, washed to remove unconjugated proteins, counted in a Neubauer chamber and stored at 4 °C in PBS containing 1 % BSA and 5 % glycerol. Streptavidin-conjugated Yellow-Green latex beads (2 µm diameter, $5 \times 10^8$ beads/preparation, Polysciences) were washed with 1 % BSA in PBS and incubated with Biotin-SP (long spacer)-conjugated Fab fragments of goat-anti-mouse IgG (H + L) (40 µg of biotinylated antibody/mg of beads, Jackson ImmunoResearch Laboratories) for 30 min at 4 °C, washed, counted in a Neubauer chamber and stored at 4 °C in PBS containing 1 % BSA and 5 % glycerol.

## Antigen-coated planar lipid bilayers (PLB)

PLB were prepared as previously described (*Dustin et al., 2007*; *Liu et al., 2012*; *Spillane and Tolar, 2017*). Briefly, liposomes were generated from 5 mM 1,2-dioleoyl-sn-glycero-3-phosphocholine plus 1,2-dioleoyl-sn-glycero-3-phosphoethanolamine-cap-biotin (Avanti Polar Lipids) at a 100:1 molar ratio by sonication. Eight-well coverslip chambers (Lab-Tek) were incubated with liposomes for 20 min at room temperature and washed with PBS. The chambers were then incubated with 1 µg/ml streptavidin (Jackson ImmunoResearch Laboratories) for 10 min, washed, and incubated with 10 µg/ml mono-biotinylated Fab' goat-anti-IgM+ G (αM-PLB) (*Liu et al., 2012*) or the same molar amount of biotinylated Tf (16 µg/ml, Sigma-Aldrich) (Tf-PLB) for 10 min at room temperature.

## COS-7 cells expressing membrane hen egg lysozyme-GFP (MHEL-GFP)

COS-7 cells were transiently transfected with mHEL-GFP (*Batista et al., 2001*) (plasmid kindly provided by Dr. Michael Gold, University of British Columbia) using Lipofectamine 3000 (Thermo Fisher Scientific) and a published protocol (*Wang et al., 2018a*), and used for experiments 24 hr post-transfection.

## Flow cytometry analysis of PM permeabilization

Mouse splenic B-cells were incubated with beads coated with αM, HEL, DEL-I or Tf in DMEM containing 6 mg/ml BSA (DMEM-BSA) at a cell:bead ratio of 1:2 (or as indicated), or with soluble F(ab')₂ goat-anti-mouse IgM+ G (sαM, 0.5 µg/ml) for 30 min at 37 °C with 5 % $CO_2$. Propidium iodide (PI, Sigma-Aldrich) was present during the 37 °C incubation as an indicator of PM permeabilization. Cells were then analyzed by flow cytometry (BD FACSCanto II) at 10,000 cell counts/sample. Bead-bound cells were identified based on their forward- (FSC) and side-scatter (SSC) properties and on fluorescence intensity (FI) when using fluorescent beads (*Figure 1—figure supplement 2*). The percentages of PI-positive (PI+) cells among the bead-bound cell populations were quantified using FlowJo 10.1 software.

## Live cell imaging of PM permeabilization

To assess PM permeabilization by protein-coated beads, mouse splenic B-cells or a B-cell line (A20) were incubated for 30 min at 4 °C in 35 mm glass-bottom dishes (MatTek) coated with poly-lysine and then with protein-coated beads at a cell:bead ratio of 1:2 for another 30 min at 4 °C. Cells were washed with DMEM-BSA and imaged in a Live Cell System chamber (Pathology Devices) at 37 °C with 5 % $CO_2$ in the presence of 50 µg/ml PI (Sigma-Aldrich) with or without 50 µM blebbistatin (Sigma-Aldrich). Images were acquired for 60 min at one frame/15–30 s using a spinning disk confocal microscope

(UltraVIEW VoX, PerkinElmer with a 63 × 1.4 N.A. oil objective). Images were analyzed using Volocity Suite (PerkinElmer) and NIH ImageJ. More than 200 cells from three independent experiments were analyzed for each condition.

To assess PM permeabilization after binding to ligand-coated PLB, splenic B-cells were incubated with FM1-43FX or FM4-64FX (Thermo Fisher Scientific) in DMEM-BSA for 5 min at 4 °C, added to coverslip chambers containing mono-biotinylated Fab' goat-anti-IgM+ G or biotinylated Tf tethered to PLB, and imaged immediately at 37 °C with 5 % $CO_2$ using a spinning disk confocal microscope (UltraVIEW VoX, PerkinElmer with a 63 × 1.4 N.A. oil objective) with or without 50 µg/ml PI and/or 10 µg/ml FM1-43FX or FM4-64FX (Thermo Fisher Scientific). Images were acquired at one frame/6–10 s and analyzed using Volocity (PerkinElmer) and NIH ImageJ. For quantitative analysis, the mean fluorescence intensity (MFI) of FM1-43FX or FM4-64FX in a defined area was measured using Volocity (PerkinElmer). More than 270 cells from three independent experiments were analyzed for each condition. For 4 hr videos, DMEM without phenol red containing 2 % FBS was used, and images were acquired at one frame/30 s in the presence of PI (50 µg/ml).

PM permeabilization was also assessed using Ponceau 4R-mediated quenching of a cytosolic fluorescent dye. B-cells were pre-stained with 1 µM CFSE (Thermo Fisher Scientific) for 10 min at 37 °C, washed with DMEM, incubated with αM- or Tf -PLB and analyzed in a spinning disk confocal microscope (UltraVIEW VoX, PerkinElmer with a 40 × 1.4 N.A. oil objective) in the presence or absence of 1 mM Ponceau 4 R (Sigma-Aldrich). More than 480 cells from four independent experiments were analyzed for each condition. To validate this method, cells pre-stained with CFSE were incubated with or without 800 ng/ml SLO in the presence or absence of 1 mM Ponceau 4 R (Sigma-Aldrich) for 10 min and analyzed by flow cytometry (BD FACSCanto II) at 10,000 cell counts/sample.

To assess the ability of antigen exposed on the surface of mammalian cells to permeabilize B-cells, COS-7 cells mock-transfected or transfected with mHEL-GFP were seeded on fibronectin-coated coverslips and cultured for 24 hr. WT or MD4 B-cells pre-stained with AF674-conjugated Fab fragments of donkey-anti–mouse IgM+ G (Jackson ImmunoResearch Laboratories) were then added to the COS-7 cells in the presence of 50 µg/ml PI and imaged immediately at 37 °C with 5 % $CO_2$ using a spinning disk confocal microscope (UltraVIEW VoX, PerkinElmer with a 40 × 1.3 N.A. oil objective). Images were acquired at one frame/20 s and analyzed using NIH ImageJ software. More than 240 cells from three independent experiments were analyzed for each condition.

## Cleaved caspase-3 detection

Splenic B-cells were pretreated or not with 1 µM staurosporine (Abcam) for 24 hr at 37 °C in DMEM-BSA to induce apoptosis (*Diaz et al., 2004*), exposed to αM- or Tf-beads for 30 min at 37 °C, washed, fixed with 4 % paraformaldehyde (PFA), blocked with 1 % BSA, and permeabilized with 0.05 % saponin. Cells were then incubated with antibodies specific for cleaved caspase-3 (Asp175) (Cell Signaling Technology) followed by AF488 donkey-anti-rabbit IgG (Life Technologies) and analyzed by flow cytometry (BD FACSCanto II) at 10,000 cell counts/sample. The percentages of cells with cleaved caspase-3 staining were determined using FlowJo 10.1 software.

## BCR signaling

BCR signaling was analyzed using both flow cytometry and western blotting. For flow cytometry assays, splenic B-cells from MD4 mice were pretreated or not with 5 µM of the Src kinase inhibitor PP2 (Millipore) (*Cheng et al., 2001*) for 30 min at 37 °C (conditions selected not to cause B-cell toxicity) and then incubated with HEL-beads in the presence or not of the inhibitor at 37 °C for 30 min. Cells were fixed with 4 % PFA, permeabilized with 0.05 % saponin, incubated with mouse anti-phosphotyrosine mAb (4G10, Millipore) followed by AF488-goat-anti-mouse $IgG_{2b}$ (Thermo Fisher Scientific) secondary antibodies, and analyzed by flow cytometry (BD FACSCanto II) at 10,000 cell counts/sample. The data were analyzed using FlowJo 10.1 software.

For western blot assays, splenic B-cells from MD4 mice were pretreated or not with 10 nM of the BTK inhibitor AVL-292 (Selleckchem) (*Aalipour and Advani, 2013*) for 30 min at 37 °C (conditions selected not to cause B-cell toxicity) and incubated with HEL-beads in the presence or not of the inhibitor at 37 °C for 30 min. Cells were then lysed using RIPA buffer (150 mM NaCl2, 1 % of NP40, 0.5 % Sodium deoxycholate, 0.1 % SDS, 50 mM Tris, pH 8.0) containing protease and phosphatase inhibitors (50 mM NaF, 1 mM $Na_3VO_4$ and 10 mM $Na_4P_2O_7$) at 4 °C. Cell lysates were run in 4–20% gradient

SDS-PAGE gels (Bio-Rad) ($5 \times 10^6$ cells/ lane) and transferred (Bio-Rad Trans-Blot transfer system) to PVDF membranes (Millipore). The membranes were blotted with rabbit anti-phospho-BTK (pBTK; Abcam) or anti-BTK (Cell Signaling Technology) antibodies followed by HRP-conjugated anti-rabbit antibodies (Jackson ImmunoResearch Laboratories) and visualization using ECL substrate (Bio-Rad) and imaging (iBright FL-1500), (Thermo Fisher Scientific).

To check if signaling affected PM permeabilization, splenic B-cells from MD4 mice were pretreated or not with 5 µM PP2 (*Cheng et al., 2001*) or 10 nM AVL-292 (*Aalipour and Advani, 2013*) for 30 min at 37 °C and then incubated with HEL-beads in the presence or not of the inhibitor and 50 µg/ml PI (Sigma-Aldrich) at 37 °C for 30 min. The percentage of PI+ cells was expressed relative to the untreated condition.

## BCR and NMII polarization

BCRs on the surface of mouse splenic B-cells were stained with Cy3-Fab donkey-anti–mouse IgM+ G (Jackson ImmunoResearch Laboratories) for 30 min at 4 °C. Cells were then incubated with αM- or Tf-beads at 4 °C for 30 min and 37 °C for different lengths of time. Cells were fixed with 4 % PFA, permeabilized with 0.05 % saponin, and incubated with rabbit anti-phosphorylated myosin light chain 2 (pMLC2) antibodies (Cell Signaling Technology) to label activated NMII (*Bresnick, 1999*), followed by AF633-goat-anti-rabbit IgG (Invitrogen). Cells were analyzed by confocal fluorescence microscopy (Zeiss LSM710 with a 63 × 1.4 N.A. oil objective). The percentages of cells with polarization of surface labeled BCRs and activated NMII towards bead-binding sites were quantified by visual inspection. More than 300 cells from three independent experiments were analyzed for each condition.

## PM repair assays

Mouse splenic B-cells were pretreated or not with 12 µM bromoenol lactone (BEL, Sigma-Aldrich) in DMEM-BSA for 30 min at 37 °C before and during assays, to inhibit lysosomal exocytosis and PM repair (*Fensome-Green et al., 2007*). Cells were then incubated with αM-beads (1:2 cell:bead ratio) with or without inhibitors at 4 °C for 5 min and 37 °C for 30 min in the presence of FM4-64FX (Thermo Fisher Scientific) to stain wounded cells. Cells were then incubated with SYTOX Blue nucleic acid stain (300 nM, Invitrogen) at 4 °C for 10 min to stain cells that failed to repair PM wounds during the 30 min incubation. Cells were analyzed by flow cytometry (BD FACSCanto II) at 10,000 cell counts/sample. Cells that were FM4-64FX positive but SYTOX Blue negative were identified as permeabilized cells that resealed. The percentages of resealed cells among all bead-bound permeabilized cells were quantified using FlowJo 10.1 software.

To assess the resealing capacity of B-cells permeabilized by ligand-coated PLB using live cell imaging, splenic B-cells were incubated with SYTOX Green (Thermo Fisher Scientific) in DMEM-BSA for 5 min at 4 °C and added to coverslip chambers containing mono-biotinylated Fab' goat-anti-IgM+ G or biotinylated Tf tethered to PLB. Cells were imaged at one frame/30 s for 4 hr at 37 °C with 5 % $CO_2$ using a spinning disk confocal microscope (UltraVIEW VoX, PerkinElmer with a 63 × 1.4 N.A. oil objective), followed by addition of 50 µg/ml PI (Thermo Fisher Scientific) at the end of the assay and final image acquisition.

## BCR polarization in relation to permeabilization

Surface BCRs of splenic B-cells were labeled with Cy5-Fab donkey-anti mouse IgG (Jackson ImmunoResearch) at 4 °C for 30 min. Cells were incubated with αM-PLB in the presence of FM 4–64 FX (Thermo Fisher Scientific) and imaged immediately at 37 °C with 5 % $CO_2$ using a spinning disk confocal microscope (UltraVIEW VoX, PerkinElmer with a 60 × 1.4 N.A. oil objective). Images were acquired at one frame/20 s for 60 min and analyzed using a custom-made MATLAB script (MathWorks) and NIH ImageJ software. BCR polarization was analyzed using maximal projection of XZ images and quantified by the MFI ratio between defined regions within the bottom half (closer to PLB) and the top half (away from PLB) of individual cells. Cells with bottom to top ratios ≥ 2 were considered polarized. More than 20 cells from three independent experiments were analyzed.

## Lysosome exocytosis

To detect LIMP-2 exposed on the cell surface, splenic B-cells (C57BL/6 or MD4) were incubated with αM-, HEL-, DEL-I or Tf-beads for 30 min at 37 °C, cooled to 4 °C, and incubated with rabbit-anti-LIMP-2

antibodies (Sigma-Aldrich) for 60 min at 4 °C. Cells were then washed and fixed with 4 % PFA, washed, blocked with 1 % BSA in PBS and incubated with AF488 donkey-anti-rabbit IgG (Life Technologies) secondary antibodies. For intracellular LIMP-2 staining, B-cells were fixed with 4 % PFA, washed, permeabilized with 0.05 % saponin for 20 min, and incubated with rabbit anti-LIMP-2 antibodies followed by AF488 donkey-anti-rabbit IgG. Flow cytometry (BD FACSCanto II) was performed at 10,000 cell counts/sample. Cells were also analyzed by confocal fluorescence microscopy (Leica SPX5 with a 63 × 1.4 N.A. oil objective). Polarization of LIMP-2 toward bound beads was quantified by calculating the fluorescence intensity ratio (FIR) of anti-LIMP-2 at the B-cell-bead contact site relative to the opposite side of the cell PM, using NIH ImageJ and a custom-made MATLAB script (MathWorks).

Individual events of lysosome exocytosis were captured using total internal reflection fluorescence (TIRF). Splenic B-cells were preloaded with SiR-Lysosome (1 µM, Cytoskeleton) in the presence of verapamil (10 µM, Cytoskeleton) for 30 min at 37 °C. Cells were added to coverslip chambers containing mono-biotinylated Fab' goat anti-IgM+ G tethered to PLB and imaged at 37 °C with 5 % $CO_2$ in the presence of PI (50 µg/ml, Sigma-Aldrich) using a TIRF microscope (NIKON Eclipse Ti-E TIRF, 63 × 1.49 NA oil objective). Images were acquired at eight frames/s during 15–20 min intervals of the 45 min incubation and analyzed using NIH ImageJ and Nikon NIS Elements software. Increases in the FI of individual SiR-Lysosome puncta (reflecting lysosome movement within the TIRF evanescent field toward the PM in contact with PLB) followed by sharp decreases within a period of 1–2 s (corresponding to a loss of the SiR-Lysosome signal upon PM fusion) were scored as exocytosis events (Jaiswal et al., 2002). More than 20 cells were analyzed in four independent experiments.

## FM endocytosis after BCR crosslinking

Mouse splenic B-cells were incubated with F(ab')₂ goat-anti-mouse IgM+ G (10 µg/ml, Jackson Immuno-noResearch Laboratories) for 10 min, followed by AF674-conjugated donkey-anti-goat (10 µg/ml, Invitrogen) for 30 min at 4 °C in coverslip chambers, to label and crosslink surface BCRs. FM1-43FX (10 µg/ml, Thermo Fisher Scientific) was added at the last 5 min of the 30 min incubation at 4 °C. Cells were washed and imaged at 37 °C with 5 % $CO_2$ in the presence of 50 µg/ml PI and 10 µg/ml FM1-43FX using a spinning disk confocal microscope (UltraVIEW VoX, PerkinElmer with a 63 × 1.4 N.A. oil objective). Images were acquired at one frame/30 s for 60 min and analyzed using Volocity (PerkinElmer).

## Assessment of BEL toxicity

Mouse splenic B-cells were pre-treated or not with 12 µM bromoenol lactone (BEL, Sigma-Aldrich) in DMEM-BSA for 30 min at 37 °C and then incubated with Tf-beads (1:2 cell-bead ratio) with or without the inhibitors at 37 °C for 30 min in the presence of SYTOX Blue (300 nM, Invitrogen). Cells were analyzed by flow cytometry (BD FACSCanto II) at 10,000 cell counts/sample. Bead-bound cells and SYTOX-Blue-positive cells were gated. The percentages of SYTOX Blue positive cells among all bead-bound permeabilized cells were quantified using FlowJo 10.1 software.

## Antigen internalization

For live imaging of antigen internalization, splenic B-cells were incubated with AF488-αM-beads (1:4 cell:bead ratio) in the presence of 1 µM SiR-Lysosome and 10 µM verapamil for 30 min at 4 °C, washed with DMEM-BSA and imaged by confocal fluorescence microscopy (Leica SPX5 with a 63 × 1.4 N.A. oil objective) for 60 min at one frame/min at 37 °C. Live time-lapse images were analyzed using NIH ImageJ.

For fixed cell imaging, splenic B-cells were pretreated or not with 50 µM blebbistatin on poly-lysine coated slides for 30 min at 4 °C and incubated with AF488-αM beads or AF488-Tf-beads at 37 °C for varying lengths of time in the presence or not of 50 µM blebbistatin. After fixation with 4 % PFA, cells were imaged by confocal fluorescence microscopy (Zeiss LSM710 with a 63 × 1.4 N.A. oil objective). Percentages of cells with intracellularly located AF488-αM puncta were determined by visual inspection of images. More than 200 cells from three independent experiments were analyzed for each condition.

For live imaging of B-cells interacting with PLB, mouse splenic B-cells were added to coverslip chambers containing PLB coated with AF488-conjugated mono-biotinylated Fab' goat-anti-mouse IgM+ G and incubated at 37 °C with 5 % $CO_2$ in the presence of 10 µg/ml FM 4–64 FX (Thermo

Fisher Scientific) for varying lengths of time. Samples were then moved to 4 °C for 5 min and immediately imaged using a confocal microscope (Leica SPX5 with a 63 × 1.4 N.A. oil objective). Internalization of antigen was quantified by determining the percentages of cells with intracellularly-located AF488-Fab' goat-anti-mouse IgM+ G puncta in each field and by measuring the AF488 FI associated with intracellular puncta in individual cells, using a custom-made MATLAB (MathWorks) script. Cells with high FM staining were identified as wounded and those with low FM staining as unwounded. More than 15 fields or ~90 cells from three independent experiments (high or low FM staining) were analyzed for each condition.

## Antigen presentation and T-cell activation

To detect antigen presentation to T-cells, splenic B-cells from F1 mice of a crossing between B10. BR-*H2$^{k2}$ H2-T18$^a$*/SgSnJJrep and MD4 mice were co-cultured with 3A9 T-cell hybridoma cells (ATCC CRL-3293) at equal concentrations ($3.75 \times 10^6$ cells/ml). Cells were incubated in DMEM supplemented with 5 % FBS and 0.05 mM 2-mercaptoethanol for 72 hr in the presence or not of soluble HEL or DEL-I (10 µg/ml), or of beads coated with HEL, DEL-I or Tf (1:4 cell: bead ratio). After incubation, the concentration of IL-2 in the supernatant was measured using an IL-2 ELISA kit (Biolegend).

## Statistical analysis

Statistical significance was assessed using unpaired, two-tailed Student's *t*-tests (Prism - GraphPad software) when only two groups were compared, and one-way ANOVA (parametric) or Kruskal-Wallis (non-parametric) when three or more groups were compared. All data were presented as the mean ± SD (standard deviation).

## Acknowledgements

We thank Dr A Upadhyaya (Department of Physics, University of Maryland) for TIRF microscopy equipment and advice, A Beaven (CBMG Imaging Core, University of Maryland) and K Class (CBMG Flow Cytometry Core, University of Maryland) for assistance with confocal microscopy and flow cytometry, Drs. S K Pierce and M Akkaya (NIH) for the 3A9 T-cell hybridoma, Dr. B Mittra and J Jensen (University of Maryland) for SLO expression and purification, Dr. Michael Gold for providing the mHEL-GFP DNA construct, and members of the Song and Andrews laboratories for helpful discussions. This work was supported by the NIH grant R01 GM064625 to NWA and WS and NIH grant T32 GM080201 to JJHvH.

# Additional information

### Funding

| Funder | Grant reference number | Author |
|---|---|---|
| National Institutes of Health | R01 GM064625 | Norma W Andrews Wenxia Song |
| National Institutes of Health | T32 GM080201 | Jurriaan JH van Haaren |

The funders had no role in study design, data collection and interpretation, or the decision to submit the work for publication.

### Author contributions

Fernando Y Maeda, Conceptualization, Formal analysis, Investigation, Methodology, Writing – original draft, Writing – review and editing; Jurriaan JH van Haaren, Conceptualization, Formal analysis, Investigation, Methodology, Software, Writing – original draft, Writing – review and editing; David B Langley, Daniel Christ, Resources, Writing – review and editing; Norma W Andrews, Conceptualization, Funding acquisition, Project administration, Supervision, Writing – review and editing; Wenxia Song, Conceptualization, Funding acquisition, Project administration, Supervision, Writing – original draft, Writing – review and editing

## Author ORCIDs
Norma W Andrews ![ORCID] http://orcid.org/0000-0002-0611-2412
Wenxia Song ![ORCID] http://orcid.org/0000-0001-8795-8657

## Ethics
This study was performed in strict accordance with the recommendations in the Guide for the Care and Use of Laboratory Animals of the National Institutes of Health. All of the animals were handled according to approved institutional animal care and use committee (IACUC) protocols (#R-JAN-18-02) of the University of Maryland. The protocol was approved by the Committee on the Ethics of Animal Experiments of the University of Maryland on January 11, 2018 .

## Decision letter and Author response
Decision letter https://doi.org/10.7554/eLife.66984.sa1
Author response https://doi.org/10.7554/eLife.66984.sa2

---

## Additional files

### Supplementary files
• Transparent reporting form

• Source code 1. iBTK inhibits BTK phosphorylation in activated B-cells.
 Western blot analysis of pBTK (A) and BTK (B, regular exposure image; C, overexposed image) in mouse splenic B-cells incubated with HEL-beads in the presence or absence of a BTK inhibitor (iBTK) for 30 min.

• Source code 2. iBTK inhibits BTK phosphorylation in activated B-cells.
 Western blot analysis of pBTK (A) and BTK (B, regular exposure image; C, overexposed image) in mouse splenic B-cells incubated with HEL-beads in the presence or absence of a BTK inhibitor (iBTK) for 30 min.

• Source code 3. iBTK inhibits BTK phosphorylation in activated B-cells.

### Data availability
All data generated or analysed during this study are included in the manuscript and supporting files.

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
