## [Editor Report]

The revisions have addressed all reviewer concerns including recovery of B cells that had undergone significant morphological change consistent with extensive plasma membrane permeabilization/lysis. Congratulations on the exciting study revealing an important role of plasma membrane permeabilization in antigen capture by B cells.

---

## [Decision Letter]

**Decision letter after peer review:**

Thank you for submitting your article "Surface-bound antigen induces B-cell permeabilization and repair facilitating antigen uptake and presentation to T-cells" for consideration by *eLife*. Your article has been reviewed by 3 peer reviewers, including Michael L Dustin as the Reviewing Editor and Reviewer #1, and the evaluation has been overseen by Suzanne Pfeffer as the Senior Editor. The following individual involved in review of your submission has agreed to reveal their identity: Andreas Mayer (Reviewer #3).

Summary:

Maeda et al. report that B cell plasma membrane permeabilization following interaction with immobilized antigens triggers a lysosome-mediated plasma membrane resealing that leads to extracellular hydrolase release and facilitates antigen capture and presentation.

Strengths of this manuscript: The questions addressed are timely and interesting. Experiments are well conceived and technically well performed. Results are largely convincing. Images and videos are of high quality.

Weaknesses of this manuscript: While the role of B cell lysosomes in antigen capture is convincingly demonstrated and extends previously reported findings, the role of B cell membrane permeabilization in triggering lysosome secretion and in initiating antigen capture is not yet fully demonstrated and may be bolstered by some suggested additional controls. There were also some concerns about potential phototoxicity in the time lapse imaging.

Essential revisions:

1) Are the cells that undergo dramatic π uptake in the time lapse videos the same cells that are competent for antigen presentation? In these sequences, the cells with high π and FM signals appear to dramatically change volume that suggests lysis- perhaps due to a combination of membrane injury during antigen capture and/or phototoxicity from the imaging. Given that the bulk cells don't lose viability, but do take up the dyes, one solution might be to sort the π positive and negative B cells that are viable by a concurrent 7AAD test and assay for antigen presentation activity to confirm that the π or FM dyes positive cells present antigen better than negative cells.

2) – The use of FM dyes to monitor cell permeabilization is problematic since these dyes are also used to monitor vesicular trafficking during endo/exocytosis (Nat. Prot 2006;1(6):2916-21). Vesicular trafficking is expected to be triggered by BCR engagement and internalization. This point should be discussed in the revised manuscript.

To further clarify this issue, it would be important to trigger B cells with soluble anti-BCR antibodies followed by secondary fluorochrome- labelled crosslinking antibodies. Under these conditions, B cell plasma membrane should not be permeabilized. FM uptake might be limited to the area of antibody capping and exhibit localization that might not overlap with the FM staining show Figure 1 and subsequent figures. The simple experiment might allow to discriminate between permeabilization and vesicular trafficking.

Results in figure 1H are not fully convincing. The observed FM entry in B cells following stimulation with F(ab')2-anti-mouse tethered to planar lipid bilayers (PLB) is relatively slow and is compatible with a process of vesicle endocytosis. It would be important to monitor FM internalization in B cells interacting with F(ab')2-anti-mouse immobilized on beads to directly compare FM uptake kinetics with the kinetics of π entry shown in Figure in Figure 1A

3) Results obtained using BEL are interesting but not fully convincing. BEL treatment has several described effects on cells included cell death induction (https://doi.org/10.1074/jbc.M307209200). The fact that in Figure 5—figure supplement 1 cells present unaltered FCS/SSC cannot exclude the possibility that B cells might be in a cell death process and for this reason might have internalized the viability dye Sytox Blue. To address this point, it is important to test the impact of BEL on non-stimulated B cells.

---

## [Author Response]

Essential revisions:1) Are the cells that undergo dramatic PI uptake in the time lapse videos the same cells that are competent for antigen presentation? In these sequences, the cells with high PI and FM signals appear to dramatically change volume that suggests lysis- perhaps due to a combination of membrane injury during antigen capture and/or phototoxicity from the imaging.

Yes, permeabilization of B cells triggered by binding to surface-associated antigen is frequently associated with specific morphological changes, visualized as an increase in cell diameter. However, it is important to note that these changes are gradually reversed, and do not lead to cell lysis. We have observed these transient morphological changes under rapid or slow image acquisition, which suggests they are not a result of phototoxicity. To further investigate the cell viability issue, we complemented our earlier flow cytometry analysis of B cells sequentially exposed to two different membrane impermeable dyes (now Figure 6A) with live imaging (new Video 11 and Figure 6C). These new results directly show that antigen-permeabilized cells undergo a surface area expansion as they become permeable to the first dye – but these same cells subsequently reseal, excluding the second dye (new Video 11 and Figure 6C). The live imaging results confirmed what we originally reported (antigen-permeabilized cells can reseal), while significantly strengthening our conclusions.

To further document the morphological recovery of antigen-permeabilized cells, we included a new supplemental figure (new Figure 6—figure supplement 2) and a new video (new Video 12) with representative examples of how permeabilized cells gradually recover their original shape and do not lyse, even when imaged for >3 additional hours from the moment of antigen-induced permeabilization. We included a brief discussion of these findings in the revised manuscript.

As previously discussed in correspondence with Suzanne Pfeffer, we were not surprised to see these transient morphological changes in permeabilized B-cells. Prior work with different cell types revealed that Ca^2+^ influx during PM permeabilization triggers transient disassembly of the cortical actin cytoskeleton, which results in extension and/or protrusion of the PM. This change is not limited to the wound site, since elevated cytosolic Ca^2+^ rapidly propagates through the cell. After the PM is repaired and Ca^2+^ influx ceases, PM extensions retract due to reassembly of the cortical actin cytoskeleton (such reversible Ca^2+^ modulation of the actin cytoskeleton was previously described by Tim Mitchison’s lab – Charras et al. Reassembly of contractile actin cortex in cell blebs, J Cell Biol 175:477-90, 2006). In addition, the massive exocytosis that occurs in permeabilized cells has long been known to cause a transient increase in cell surface area (McNeil and Steinhardt J Cell Biol 137: 1-4, 1997), and in this case, recovery is slower since it requires sustained membrane traffic. Our imaging results suggest that both mechanisms (actin cytoskeleton reassembly and endocytic membrane traffic) are probably involved in the recovery of antigen-permeabilized B cells, and we are very interested in characterizing this process in future studies.

Given that the bulk cells don't lose viability, but do take up the dyes, one solution might be to sort the PI positive and negative B cells that are viable by a concurrent 7AAD test and assay for antigen presentation activity to confirm that the PI or FM dyes positive cells present antigen better than negative cells.

This was an excellent suggestion, since this experiment would allow us to directly link antigen-induced B-cell PM permeabilization to antigen presentation. We spent several months troubleshooting and optimizing protocols, and completed more than ten trials of the suggested sorting experiment. Unfortunately, we encountered two significant hurdles that prevented us from reaching our goal. The first hurdle was the need to avoid continued B-cell permeabilization during the 24-h incubation with T-cells, since sorting cannot remove cell-bound beads. We tried to address the issue by fixing cells immediately after sorting. However, even low fixative concentrations destroyed the T-cell recognition site in the MHCII-HEL peptide complexes presented by the primary B-cells used in our assay. We identified this problem by comparing the ability of fixed and unfixed B-cells to activate T-cells after incubation with HEL-beads for 2 h. The second hurdle we encountered was the low yield of the sorting procedure. Based on our data, with ~50% of B-cells binding HEL-beads at each given time, 15-20% of antigen-bead-bound cells are permeabilized, and ~50% of these permeabilized B-cells reseal their plasma membrane – which results in only 5% of the initial number of cells, even without considering the inevitable loss during sorting. Through our extensive trials, we learned that bead binding and incorporation of membrane impermeable dyes into the B-cell DNA significantly increased cell loss, further decreasing the sorting yield. Thus, despite numerous attempts we failed to obtain a sufficient number of sorted B-cells to perform reliable T-cell activation analysis. However, even though we could not directly confirm the relationship between antigen-induced B-cell membrane permeabilization and antigen presentation through cell sorting, our study has revealed a direct link between antigen-induced B-cell PM permeabilization and antigen internalization (Figure 7D-F), a critical step that initiates antigen presentation.

2) The use of FM dyes to monitor cell permeabilization is problematic since these dyes are also used to monitor vesicular trafficking during endo/exocytosis (Nat. Prot 2006;1(6):2916-21). Vesicular trafficking is expected to be triggered by BCR engagement and internalization. This point should be discussed in the revised manuscript.

Discussion of this point has been included in the revised manuscript (lines 156 to 166 in Results and lines 518-528 in Discussion).

We are certainly aware that lipophilic dyes can also enter B-cells through receptor endocytosis (Cousin et al. Methods Mol Biol 1847:239-49, 2018), particularly when BCR endocytosis is induced by receptor cross-linking (Song et al. J Immunol 155: 4255-63, 1995). However, under our experimental conditions (see below), endocytosed lipophilic dyes appeared as puncta that gradually accumulated at the cell periphery. This pattern was in sharp contrast to the sudden, massive influx of lipophilic dyes that occurs ~8 min before PI influx.

To further clarify this issue, it would be important to trigger B cells with soluble anti-BCR antibodies followed by secondary fluorochrome- labelled crosslinking antibodies. Under these conditions, B cell plasma membrane should not be permeabilized. FM uptake might be limited to the area of antibody capping and exhibit localization that might not overlap with the FM staining show Figure 1 and subsequent figures. The simple experiment might allow to discriminate between permeabilization and vesicular trafficking.

We performed the experiment suggested by the reviewer and included the data in the new Figure 1—figure supplement 6. We activated BCR endocytosis by cross-linking surface BCRs using soluble F(ab’)_2_ goat-anti-mouse IgM+G antibodies and fluorescent F(ab’)_2_ anti-goat-IgG. Under these conditions, which did not cause PM permeabilization, we observed FM1-43 uptake appearing as small peripheral puncta that colocalized with BCR cross-linking antibodies. Such endosome-associated FM1-43 staining pattern was markedly different from the sudden, massive FM influx observed shortly before PI entry in permeabilized cells (see a comparison of both conditions in the new Video 4). This experiment allowed us to clearly distinguish the influx of FM during B-cell permeabilization from the peripheral FM puncta formed during endocytosis – we thank the reviewer for the suggestion.

Results in figure 1H are not fully convincing. The observed FM entry in B cells following stimulation with F(ab')2-anti-mouse tethered to planar lipid bilayers (PLB) is relatively slow and is compatible with a process of vesicle endocytosis. It would be important to monitor FM internalization in B cells interacting with F(ab')2-anti-mouse immobilized on beads to directly compare FM uptake kinetics with the kinetics of π entry shown in Figure in Figure 1A

We hope that the experiment above showing the distinct, exclusively peripheral endocytosis pattern triggered by BCR cross-linking by soluble antibodies clarified this point. We also added a new panel to Figure 1 (new Figure 1I) to emphasize that FM dye influx during antigen-induced permeabilization is a sudden event, with an influx kinetics similar to what is observed with PI (more examples are provided in the new Figure 1 supplement 4). Furthermore, unlike what is seen during endocytosis, in antigen-permeabilized cells FM influx frequently results in staining of the nuclear envelope (note the juxtaposition of FM staining and PI-stained nuclei in Figure 1H and Figure 1-supplement 4A).

3) Results obtained using BEL are interesting but not fully convincing. BEL treatment has several described effects on cells included cell death induction (https://doi.org/10.1074/jbc.M307209200). The fact that in Figure 5—figure supplement 1 cells present unaltered FCS/SSC cannot exclude the possibility that B cells might be in a cell death process and for this reason might have internalized the viability dye Sytox Blue. To address this point, it is important to test the impact of BEL on non-stimulated B cells.

We agree that our data showing unaltered FCS/SSC did not fully rule out potential cell death induction by BEL. To directly examine this issue, we incubated B cells with Tf-beads (our control condition that induces very low levels of B cell permeabilization) in the presence or absence of BEL and examined the cell’s susceptibility to SYTOX Blue staining. As shown in the new Figure 6-supplement 1C,D, treatment with BEL under our experimental conditions (12 µM for 30 min) did not increase the percentage of SYTOX-permeable B-cells.